# IMPROVING GRADIENT-GUIDED NESTED SAMPLING FOR POSTERIOR INFERENCE

## ABSTRACT

We present a performant, general-purpose gradient-guided nested sampling algorithm, GGNS, combining the state of the art in differentiable programming, Hamiltonian slice sampling, clustering, mode separation, dynamic nested sampling, and parallelization. This unique combination allows GGNS to scale well with dimensionality and perform competitively on a variety of synthetic and real-world problems. We also show the potential of combining nested sampling with generative flow networks to obtain large amounts of high-quality samples from the posterior distribution. This combination leads to faster mode discovery and more accurate estimates of the partition function.

## 1 INTRODUCTION

Bayesian parameter estimation and model comparison are key to most scientific disciplines and remain challenging problems, especially in high-dimensional and multimodal settings. While traditionally Markov chain Monte Carlo (MCMC) methods have been used to perform Bayesian inference, differentiable programming has enabled the development of new, more efficient algorithms, such as variational inference (MacKay, 2003), Hamiltonian Monte Carlo (Duane et al., 1987; Neal et al., 2011), and Langevin dynamics (Besag, 1994; Roberts & Tweedie, 1996; Roberts & Rosenthal, 1998); as well as more recent learning-based methods such as the Path Integral Sampler (Zhang & Chen, 2022) and generative flow networks (Bengio et al., 2023; Lahlou et al., 2023).

From the perspective of differential programming, less attention has been paid in recent years to nested sampling (Skilling, 2006; Buchner, 2021; Ashton et al., 2022), which is a widely used algorithm for Bayesian parameter inference and model comparison. Nested sampling has been used in a range of applications in the natural sciences, from cosmology (Mukherjee et al., 2006; Handley et al., 2015a) and astrophysics (Lavie et al., 2017; Günther & Daylan, 2021) to particle physics (Yallup et al., 2022) and biology (Russel et al., 2018). Furthermore, it provides both samples from the posterior distribution and an estimate of the Bayesian evidence, which can be used for model comparison (Marshall et al., 2006) or to test compatibility between datasets (Handley & Lemos, 2019).

The key challenge in implementing a nested sampling algorithm is constructing a method that generates samples drawn from the prior, subject to a hard likelihood constraint. Whilst there is a wide variety of publicly available implementations for doing so (see Buchner, 2021; Ashton et al., 2022, for an exhaustive list), these methods are only capable of scaling to hundreds of dimensions (Handley, 2023).

One way to improve the performance of nested sampling algorithms is to use the information about the gradient of the likelihood to propose new points. However, gradient-guided sampling on constrained domains is not straightforward. Whilst the materials science and chemistry literature has made extensive use of gradient-guided nested sampling (Baldock et al., 2017; Pártay et al., 2021; Habeck, 2015; Nielsen, 2013), these methods are often generally bespoke to their physical problems of interest and are not suitable as general-purpose Bayesian samplers. Previous works, such as Betancourt (2011); Speagle (2020) have shown the potential of reflective slice sampling (Neal, 2003), also known as Galilean Monte Carlo (GMC) (Feroz & Skilling, 2013) or Hamiltonian slice sampling (HSS) (Zhang et al., 2016; Bloem-Reddy & Cunningham, 2016), for general-purpose sampling, including the case of non-smooth functions (Mohasel Afshar & Domke, 2015). However, these approaches have found HSS to be *"substantially less efficient (and in general less reliable) than other gradient-based approaches"*.[1] Another alternative is recent work in proximal nested sampling (Cai et al., 2022; McEwen et al., 2023), which uses proximal operators to propose new points. However, this method only works for log-concave likelihoods.

---

[1] Verbatim from the documentation of the dynesty algorithm.

In this work, we combine ideas from across the nested sampling literature and learning-based samplers and create a new gradient-guided nested sampling (GGNS) algorithm. The four key differences with previous work are 1) the use of self-tuning HSS in combination with gradients calculated through differentiable programming to propose new points, 2) incorporating parallel updates using ideas from dynamic nested sampling (Higson et al., 2019; Speagle, 2020; Handley et al., 2015b) to increase the speed of calculations, 3) A novel termination criterion, and 4) cluster identification to avoid mode collapse. We show that with these changes in combination, our GGNS algorithm scales to significantly higher-dimensional problems without necessitating a proportional increase in the number of live points with respect to dimensionality. This allows GGNS to perform fast and reliable inference for high-dimensional problems.

We show that the GGNS method presented in this work can be used to perform inference in a wide range of problems and that it can be used to improve the performance of existing nested sampling algorithms. Furthermore, we compare our method to existing algorithms for posterior inference and show that it outperforms them, particularly when dealing with highly multimodal distributions. One of the main advantages of the proposed approach is that it requires little hyperparameter tuning and can be used out-of-the-box in a wide range of problems.

Finally, we show the potential of combining nested sampling with generative flow networks (GFlowNets, Bengio et al., 2021; 2023), which are policy learning algorithms that are trained to generate samples from a target distribution and can flexibly be trained off-policy (Malkin et al., 2023). We show how we can use nested sampling to guide GFlowNet training, leading to faster mode finding and convergence of evidence estimates than with traditional GFlowNets. Conversely, we also show how the amortization achieved by GFlowNets can be used to obtain large amounts of high-quality samples from the posterior distribution.

## 2 BACKGROUND AND RELATED WORK

### 2.1 NESTED SAMPLING

Nested sampling is used for estimating the marginal likelihood, also known as the evidence, in Bayesian inference problems (Skilling, 2006):

$$\mathcal{Z} = \int \mathcal{L}(\theta)\pi(\theta)d\theta, \tag{1}$$

where $\mathcal{L}(\theta)$ is the likelihood function, and $\pi(\theta)$ is the prior distribution. This integral is often intractable due to the high dimensionality and complexity of modern statistical models. In the process of calculating this integral, nested sampling also produces samples from the posterior distribution.

At its core, nested sampling transforms the evidence integral into a one-dimensional nested sequence of likelihood-weighted prior mass, allowing for efficient exploration of the parameter space. The key idea is to enclose the region of high likelihood within a series of nested iso-likelihood contours. This is achieved by introducing a set of *live points* distributed within the prior space and successively updating this set by iteratively replacing the point with the lowest likelihood with a new point drawn from the prior while ensuring the likelihood remains above a likelihood threshold.

As nested sampling progresses, it adaptively refines the prior volume containing higher-likelihood regions. By constructing a sequence of increasing likelihood thresholds, nested sampling naturally focuses on the most informative regions of parameter space. Consequently, nested sampling offers several advantages, including robustness to multimodality in posterior distributions, convergence guarantees, and the ability to estimate posterior probabilities and model comparison metrics. A more detailed review of the algorithm can be found in appendix A.

The number of nested sampling likelihood evaluations scales as (Skilling, 2006; Handley, 2023):

$$n_{\text{like}} \propto n_{\text{live}} \times f_{\text{sampler}} \times \mathcal{D}_{\text{KL}}(\mathcal{P}|\Pi), \tag{2}$$

where $n_{\text{live}}$ is the number of live points, $f_{\text{sampler}}$ is the efficiency of the live point generation method (the average number of likelihood evaluations required to generate each new sample), and $\mathcal{D}_{\text{KL}}(\mathcal{P}|\Pi)$ is the Kullback-Leibler divergence between the posterior and the prior.

To understand the scaling of nested sampling with dimensionality, we should consider the three terms separately. Here, $\mathcal{D}_{\text{KL}}(\mathcal{P}|\Pi)$ is fixed by the problem at hand (so cannot be modified without substantial adjustment of the meta-algorithm (Petrosyan & Handley, 2022)), and is usually assumed to scale linearly with the number of dimensions. $n_{\text{live}}$ for most algorithms scales linearly with dimensionality for two independent reasons: First since the uncertainty in the log-evidence estimation

is approximately (Skilling, 2006)

$$\sigma(\log Z) \approx \sqrt{\mathcal{D}_{\text{KL}}(\mathcal{P}|\Pi)/n_{\text{live}}}, \tag{3}$$

if we wish to keep this constant we must scale $n_{\text{live}}$ with $\mathcal{D}_{\text{KL}}(\mathcal{P}|\Pi)$, which as discussed before scales linearly with dimension. Second, most practical live point generation methods require a minimum number of points to tune their internal parameters (such as ellipsoidal/cholesky decompositions or neural network training), and this minimum number scales with dimensionality. In the next section, we describe $f_{\text{sampler}}$ scaling.

## 2.2 PREVIOUS WORK

The key difficulty in nested sampling is that to generate a new point, one needs to sample points from the prior subject to a hard likelihood constraint:

$$\{\theta \sim \pi : \mathcal{L}(\theta) > \mathcal{L}_*\}. \tag{4}$$

Broadly, the mechanisms for achieving this fall into two classes: *region sampling* and *step sampling* (Ashton et al., 2022). Region samplers have excellent performance in low dimensions, but have a computational cost that scales exponentially with dimensionality $f_{\text{sampler}} \sim O(e^{d/d_0})$, where $d_0 \sim O(10)$ is both method and problem dependent. Step samplers have a live point generation cost that scales linearly with dimensionality $f_{\text{sampler}} \sim O(d)$, so are less efficient in low dimensions.

*Region samplers* use the current set of live points to define a proxy that encapsulates the likelihood-constrained region eq. (4), and then appropriately samples from this proxy. For example MultiNest (Feroz & Hobson, 2008; Feroz et al., 2009; Feroz et al., 2019) achieves this with an ellipsoidal decompsition fit to the current set of live points, nessai (Williams et al., 2021; 2023) trains a normalising flow and ultranest (Buchner, 2021) places ellipsoidal kernels on each live point.

*Step samplers* run a Markov chain starting from one of the current live points, terminating when one has decorrelated from the initial point and then using the final point of the chain as new point. Whilst Skilling (Skilling, 2006) originally envisaged a Metropolis Hastings step, in practice on its own this is a poor choice for sampling from hard-bounded regions. proxnest (Cai et al., 2022; McEwen et al., 2023) uses prox-guided Langevin diffusion, DNest (Brewer & Foreman-Mackey, 2016) offers a flexible framework for programming one's own stepper, neuralnest (Moss, 2020) uses normalizing flow guided Metropolis steps and PolyChord (Handley et al., 2015a;b) uses slice sampling. Finally, dynesty (Speagle, 2020) and ultranest (Buchner, 2021) offer Python re-implementations of many of the above within a single package, with a default dimensionality-dependent switching between region and path sampling.

Dynamic nested sampling (Higson et al., 2019; Speagle, 2020) is a variant of nested sampling which proposes eliminating and replacing multiple points at each iteration. It was initially implemented in the dyPolyChord[2] and dynesty[3] packages, but now is common to many implementations (Ashton et al., 2022). It has two main use-cases; increasing the number of posterior samples generated by nested sampling, and implementing parallelization schemes.

## 2.3 HAMILTONIAN SLICE SAMPLING

HSS was first introduced in the context of slice sampling (Neal, 2003), as a variant of Hamiltonian Monte Carlo. As in slice sampling, the algorithm initially selects an *initial point* from the current set of live points and a direction. An initial *momentum* variable $p_{\text{ini}}$, which is a $d$-dimensional array (where $d$ is the dimension of the space), is also defined, typically by randomly sampling a unit vector. The algorithm then proceeds by simulating the *trajectory* of a particle located at the initial point with the chosen initial velocity integrated with some time step $\Delta t$, such that at each step the position of the particle is updated according to $x' = x + p\Delta t$. When the particle goes beyond the slice, it is *reflected* back into the slice. This reflection is performed by updating the momentum from $p$ to $p'$ using the equation

$$p' = p - 2(p \cdot n)n, \quad n := \nabla\mathcal{L}(\theta)/\|\nabla\mathcal{L}(\theta)\|, \tag{5}$$

where $n$ is the unit vector in the direction of the likelihood gradient and thus the normal vector to an iso-likelihood surface. Note that, because we are only using the direction of the gradient, one can

---

[2]https://dypolychord.readthedocs.io/en/latest/
[3]https://dynesty.readthedocs.io/en/stable/

equivalently use the gradient of the log-likelihood, i.e. the score, which is more efficient to compute. We summarize the algorithm in appendix B.

As highlighted in Neal (2003), eq. (5) is only exact when the point where the reflection of the trajectory of the particle takes place is exactly on the boundary $\mathcal{L}(\theta) = \mathcal{L}_*$. In practice, we can either use a small tolerance $\epsilon$ to define a neighborhood around the slice and reflect a trajectory whenever the particle is within this neighborhood, or reflect a trajectory whenever the particle lands at a point outside the boundary. The latter method has a theoretical risk of a particle getting "stuck" behind the boundary (in which case the trajectory would be rejected, and a new initial momentum would be chosen).

HSS (or GMC) has been used for nested sampling before (Betancourt, 2011; Feroz & Skilling, 2013; Speagle, 2020). However, the `dynesty` implementation and defaults of HSS lacks the efficiency and reliability of other sampling methods. In addition, in these public implementations, a score has to be manually provided since the package is not compatible with modern differentiable programming frameworks.

## 3 CONTRIBUTIONS

In this section, we outline the key combination of ingredients in `GGNS` we use to significantly improve its performance in high-dimensional settings in comparison with existing publicly available tools.

In brief: we introduce trimming & adaptive step size techniques to remove the hyperparameter tuning difficulties that have beset previous implementations, bring in the current state-of-the-art in parallelization and cluster recognition, and implement in differentiable programming which removes the requirement of providing a score function. With these innovations we find that one only needs $\sim O(1)$ bounces to have decorrelated the chain from the start point, allowing sublinear $f_{\text{sampler}}$ scaling. Finally, for maximum posterior scaling, the fact that gradients guide the path means one no longer requires $n_{\text{live}} \sim O(d)$, giving an in-principle linear scaling with dimensionality for the purposes of posterior estimation.

This linear scaling has a theoretical basis. For methods such as slice sampling, taking $n$ steps in a $d$ dimensional space leads to sampling an $n$-dimensional subspace. Therefore, we need to reach $O(d)$ steps to explore the full space. For Hamiltonian slice sampling, on the other hand, every time we use the gradient for a reflection, we get information about the full $d$ dimensional space. Therefore, each step is exploring the full volume, leading to the requirement of $O(1)$ reflections. This is a similar argument to the better scaling with the dimensionality of Hamiltonian Monte Carlo methods, compared to methods such as random walk Metropolis-Hastings.

In detail, our contributions are the following. We include a complete algorithm in Appendix H ablation studies showing the importance of various components in Appendix G.

**Adaptive Time Step Control**     We add an adaptive time-step control mechanism in the HSS algorithm. In HSS, particles move in straight lines and eventually reflect off the hard likelihood boundary. To ensure the trajectories between reflections strike a balance between efficiency and accuracy, we introduce the concept of a variable time step, denoted as d$t$. This time step is adjusted dynamically during the course of the algorithm. By monitoring the number of reflections, we increase or decrease d$t$ to optimize the computational efficiency while maintaining trajectory integrity. This approach, inspired by Neal's work in (Neal, 2003), enables us to employ larger time steps, thereby reducing the number of reflections without compromising trajectory quality.

**Trajectory Preservation**     In our second enhancement, we introduce a novel approach to preserving and utilizing trajectory[4] information during the HSS updates. Specifically, we store all points along the trajectory after a designated number of reflections, where `min_ref < max_ref`. This archive of trajectory points allows us to efficiently select a new live point by uniformly sampling the stored trajectories in a fully parallel manner. We also perturb trajectories by adding some noise `delta_p`, to achieve faster decorrelation of the samples.

**Pruning Mechanism**     To further enhance efficiency, we introduce a "pruning" mechanism during the HSS process. Points that have remained outside the slice for an extended duration are identified and removed from consideration. These pruned points are then reset to their initial positions, and new momenta are randomly assigned. This mechanism significantly improves the computa-

---

[4]The term trajectory here refers to the states of the chain, not the intermediate states of a Hamiltonian trajectory, as in Nishimura & Dunson (2020).

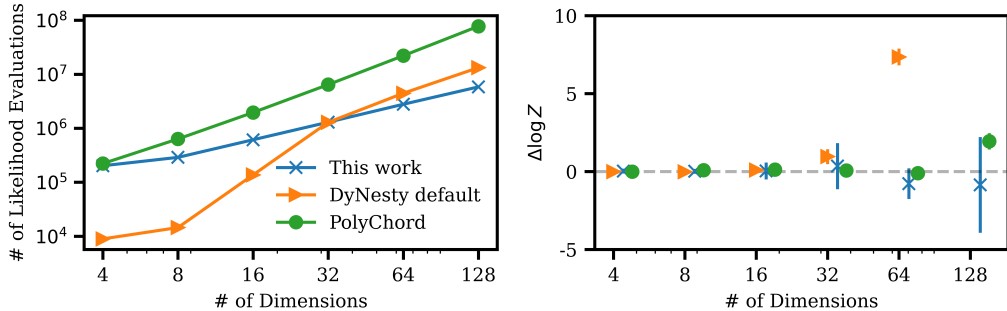

Figure 1: Comparison of likelihood evaluations (left panel) and error in the estimation of $\log \mathcal{Z}$ for different dimensionalities between this work (blue), and other nested sampling algorithms (`PolyChord` in green and `dynesty` in orange; showing more efficient log-log linear scaling while achieving a higher-fidelity estimate. All comparisons are done for a Gaussian likelihood with a diagonal covariance matrix. The error bars show the standard deviation over 10 runs. Error bars for `PolyChord` and `dynesty` are also present, but barely visible. Note that the last point for `dynesty` is not shown, as it is too large to fit in the plot.

tional efficiency of the proposed method, as we do not waste computational resources evaluating the likelihood of points that have drifted far away from the slice.

**Parallel Evolution of Live Points** As in Burkoff et al. (2012); Henderson & Goggans (2014); Martiniani et al. (2014); Handley et al. (2015b), we implement a dynamic approach to live point management, whereby half of the live points are "killed" at each iteration and replaced with new points. The new set of live points evolves with our HSS algorithm entirely in parallel, given that the HSS algorithm boils down to simulating simple dynamics for all the live points. This parallelism dramatically accelerates the algorithm's execution.

**Mode Collapse Mitigation** To address the issue of mode collapse, we incorporate a clustering recognition and evolution algorithm as developed and implemented in `PolyChord` (Handley et al., 2015b). During the execution of the nested sampling process, we identify clusters of points and keep track of the volume of each cluster. Then, we spawn points proportionally to this volume. This addition helps maintain diversity among live points, preventing them from converging prematurely to a single mode.

**Robust Termination Criterion** Our final contribution involves the introduction of an alternative termination criterion, which we find to more robust. Unlike previous implementations of nested sampling that rely on the remaining prior volume $X$, we utilize the property that the quantity $X\mathcal{L}(\theta)$ follows a characteristic trajectory—initially increasing, reaching a peak, and then decreasing. We terminate the algorithm when $X\mathcal{L}(\theta)$ has decreased by a predetermined fraction from its maximum value. This termination is further explained in appendix I. This criterion proves to be more resilient to variations in hyperparameters, including the number of live points.

**Differentiable Programming** Whilst nested sampling algorithms written in differential programming languages exist in `jax` (Albert, 2020) and `torch` (Paszke et al., 2019; Anau Montel et al., 2023), these do not make use of gradients in guiding the choice of a new live point, Therefore, their choice of using a differentiable programming language is motivated mainly by the advantages of GPU interoperability. To our knowledge, ours is the first algorithm utilizing the gradients derived by differentiable programming to guide the choice of a new live point. Furthermore, this adaptation of nested sampling to hardware intended for modern machine learning workflows, featuring massive parallelization on GPUs; is particularly important in data processing settings that combine nested sampling with deep learning, such as when the prior or likelihood models are given by deep neural networks. We show an example of this when we combine nested sampling and generative flow networks in Section 5.

We summarize the hyperparameters in appendix C and provide an ablation study in appendix G.

Table 1: Log-evidence function estimation bias (mean and standard deviation over 10 runs). The first rows are from our method, while the rest are from Zhang & Chen (2022); Lahlou et al. (2023). Note that the last three methods are using importance-weighted bound $B_{\text{RW}}$. In bold font, we show the estimates that are unbiased at the one standard deviation level.

| Method | Gaussian mixture | Funnel |
|---|---|---|
| HMC | $-1.876 \pm 0.527$ | $-0.835 \pm 0.257$ |
| SMC | $-0.362 \pm 0.293$ | $-0.216 \pm 0.157$ |
| On-policy PIS-NN | $-1.192 \pm 0.482$ | $\mathbf{-0.018 \pm 0.020}$ |
| Off-policy GFlowNet TB | $\mathbf{-0.003 \pm 0.011}$ | $-0.026 \pm 0.020$ |
| On-policy GFlowNet TB | $-1.301 \pm 0.434$ | $\mathbf{-0.012 \pm 0.108}$ |
| Ours | $\mathbf{0.029 \pm 0.132}$ | $\mathbf{-0.051 \pm 0.353}$ |

## 4 EXPERIMENTS

### 4.1 COMPARISON WITH OTHER NESTED SAMPLING METHODS

We compare the performance of gradient-guided nested sampling with two popular nested sampling algorithms, already introduced in Section 2: PolyChord and dynesty. We use the same likelihood function for all algorithms, which is a Gaussian likelihood with a diagonal covariance matrix, and therefore has $\mathcal{D}_{\text{KL}}(\mathcal{P}|\Pi) \propto d$.

For PolyChord, since $f_{\text{sampler}} \propto n_{\text{repeats}} = 5d$, from eq. (2) we therefore expect $n_{\text{like}} \propto n_{\text{live}} \times 5d^2$. For dynesty, its default mode swaps between a region sampler with $n_{\text{like}} \propto n_{\text{live}} \times e^{d/d_0}d$ in low dimensions to a slice sampler with $n_{\text{repeats}} = d$, giving $n_{\text{like}} \propto n_{\text{live}} \times d^2$. For GGNS, since $f_{\text{sampler}} \sim \text{max\_ref} \sim O(1)$, we instead expect $n_{\text{like}} \propto n_{\text{live}} \times d$.

For demonstrating the various competing effects discussed in Sections 2.1, 2.2 and 3, we set $n_{live} = 200$, independent of dimensionality. Note that constant $n_{\text{live}}$ mode is not usually recommended for these samplers, since as discussed in Section 2.1 we need a minimum number of live points to tune the live point generation hyperparameters. Since GGNS uses gradients to guide the choice of live points, it is not restricted in this way.

The results are shown in Figure 1. We observe the scaling expected from the discussion above. At constant $n_{\text{live}}$, PolyChord has quadratic scaling with dimensionality, providing good evidence estimates until the dimensionality becomes similar to the $n_{\text{live}} = 200$. dynesty is most efficient but exponentially scaling in low dimensions, and swaps to quadratic scaling in higher dimensions when it moves over to slice sampling, at a lower constant than PolyChord due to its default $n_{\text{repeats}} = d$ in comparison with $5d$. Note however that this factor of 5 default efficiency is traded off against poor evidence estimates, even in low dimensions, once it is in slice sampling mode.

GGNS, as predicted, has by far the best (linear) scaling and performs evidence estimation accurately even as the dimensionality approaches the number of live points since its live point generation is guided by gradients rather than the other live points. Note, however, that as expected from eq. (3) the error increases with the square root of the dimensionality at fixed $n_{\text{live}}$.

### 4.2 CALCULATION OF EVIDENCE

The calculation of the Bayesian evidence is a good way to evaluate the performance of inference algorithms. In this section, we confirm the performance of gradient-guided nested sampling with other methods to sample from a target density. We compare with the following methods: Hamiltonian Monte Carlo (HMC, MacKay, 2003; Hoffman et al., 2014), Sequential Monte Carlo (SMC, Halton, 1962; Gordon et al., 1993; Chopin, 2002; Del Moral et al., 2006), Path Integral Sampler (PIS, Zhang & Chen, 2022) and generative flow networks (GFlowNet Bengio et al., 2021; 2023; Lahlou et al., 2023). For SMC, the settings follow the code release of Arbel et al. (2021)[5]. For PIS, we compare with the on-policy version alone, as it obtains better results than the off-policy version. For GFlowNet, we compare with the off and on policy versions, as they perform differently for different tasks, the former being better for multimodal distributions as it is better at exploration, and the latter requiring less samples to converge. We focus on GFlowNets trained with the trajectory balance loss (Malkin et al., 2022).

We compare these methods with GGNS in two tasks, already introduced in (Hoffman et al., 2014; 2019; Zhang & Chen, 2022; Lahlou et al., 2023): The first one is the **funnel distribution**, which is a 10D distribution with a funnel shape. The second one is a **Gaussian mixture** in 2-dimension, which consists of a mixture of 9 mode-separated Gaussians.

---

[5] https://github.com/deepmind/annealed_flow_transport.git.

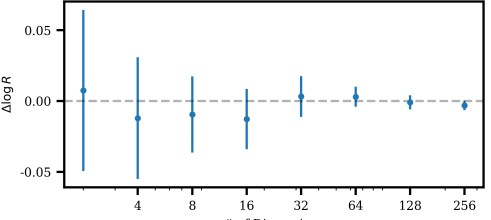

Figure 2: **First row:** The true image and noise that we aim to reconstruct. **Second row:** The mean from out gradient-guided nested sampling and the standard deviation. We see how the `GGNS` posterior matches the expected one.

Figure 3: The error in the estimate of the normalization of the reward function for the torus task, as a function of the dimensionality of the problem. We see that `GGNS` obtains consistent estimates of the normalization, even in high dimensional settings.

As our benchmark, we use the accuracy of the estimate in the log-evidence, or log-partition function. We report the mean and standard deviation of the estimation bias over 10 independent runs in Section 4.2. We observe that gradient-guided nested sampling obtains unbiased estimates in both tasks, something that does not happen for any of the other methods studied in this work. While our standard deviation is higher than that of other methods, these can be reduced by adjusting the hyperparameters of our method. However, eq. (3) shows that the nested sampling log-evidence error can only be reduced sublinearly by increasing the number of live points $n_{\text{live}}$, which increases the computational cost. We cannot therefore expect substantial improvements in `GGNS` log-evidence error bars without innovations in the nested sampling algorithm itself.

## 4.3 IMAGE GENERATION

We also tested the performance of `GGNS` on a high-dimensional problem, sampling the posterior distribution over image pixels. To do this, we chose the problem of inferring the pixel values of background galaxies in strong gravitational lensing systems (e.g., Adam et al., 2022). We assumed a correlated (and non-zero mean) normal prior distribution for the background source. A sample from the prior was generated (representing the background galaxy) and was distorted by a the potential of a foreground lens. Gaussian noise was then added to produce a noisy simulated data. Given the data, the posterior of a model (a pixelated image of the undistorted background source) could be calculated by adding the likelihood and the prior terms. Furthermore since the model is perfectly linear (and known) and the noise and the prior are Gaussian, the posterior is a high-dimensional Gaussian posterior that could be calculated analytically, allowing us to compare the samples drawn with `GGNS` with the analytic solution.

Figure 2 shows a comparison between the true image, and its noise, and the one recovered by `GGNS`. We see that we can recover both the correct image, and the noise distribution. We emphasize that this is a uni-modal problem and that the experiment's goal is to demonstrate the capability of `GGNS` to sample in high dimensions (in this case, 256), such as images, and to test the agreement between the samples and a baseline analytic solution.

## 4.4 SYNTHETIC MOLECULE TASK

Finally, we test `GGNS` on task, inspired by molecular conformations[6]. First, we build a reward function on an $n$ dimensional torus, which extends the reward function introduced in Lahlou et al. (2023) to higher dimensional spaces. We define the reward function as:

$$R_n(\mathbf{x}, \alpha, \beta, c) = \left( \sum_{i \text{ even}}^{n} \sin(\alpha x_i) + \sum_{j \text{ odd}}^{n} \cos(\beta x_j) + c \right)^3, \qquad x_i \in [0, 2\pi). \tag{6}$$

---

[6]To apply this to real molecular configurations, we need a fully differentiable chemical simulator. We leave this for future work

FAB                                        Gradient-Guided Nested Sampling

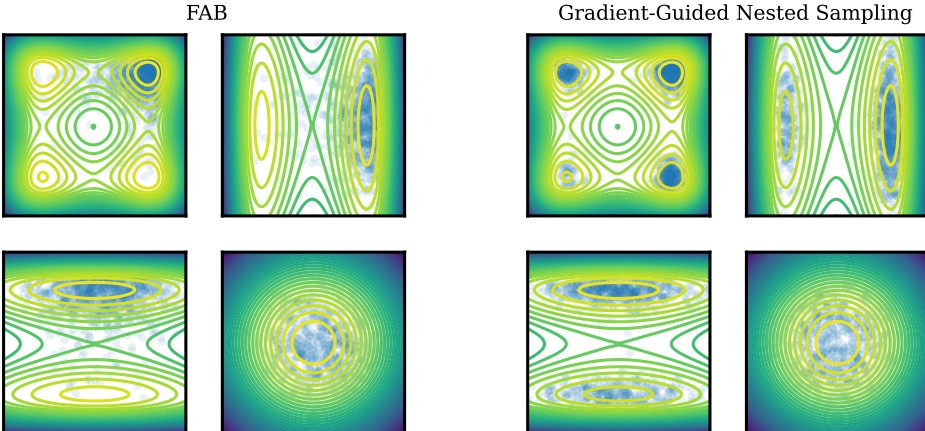

Figure 4: Contour lines for the target distribution, and samples, for the first four dimensions of the 32-dimensional "Many Wells" problem. On the left, we show results for Flow Annealed Importance Sampling Boostrap (FAB) with a replay buffer, and on the right for `GGNS`. Unlike FAB, `GGNS` recovers all modes.

This reward function models the multimodality we expect in molecular conformations, but has the advantage of having a normalization that can be calculated analytically, as detailed in appendix F. This means that we can assess the accuracy of `GGNS` by comparing the estimated normalization with the true value in high dimensional settings. The results are shown in Section 4.4, where we see that `GGNS` obtains consistent estimates of the normalization, even in high dimensional settings.

We also compare our method to Flow Annealed Importance Sampling Boostrap (FAB, Midgley et al., 2022), with a replay buffer. This method has achieved state of the art results in sampling tasks, and was already applied to the Boltzmann distribution of the alanine dipeptide molecule. We use `GGNS` in two synthetic tasks first introduced in Midgley et al. (2022): A mixture of 40 Gaussians in two dimensions, and the 32-dimensional "Many Well" problem [7]. The Many Well problem is a particularly challenging one, due to the high its high dimensionality. We show results for the first four dimensions in Figure 4. We see how `GGNS` does an even better job than FAB at recovering all existing modes. We show visual comparison for the torus reward eq. (6) and the mixture of 40 Gaussians in appendix D.

## 5 COMBINATION WITH GENERATIVE FLOW NETWORKS

We show how the samples obtained from the proposed nested sampling procedure can augment amortized sampling algorithms, such as the generative flow networks considered in Section 4.2. In Lahlou et al. (2023), it was shown that Euler-Maruyama integration of a stochastic differential equation (SDE) can be viewed as the generative process of a generative flow network. The drift and diffusion terms of the SDE can be trained as the GFlowNet's forward policy to sample from a target distribution given by an unnormalized density. In particular, GFlowNet objectives can be used to learn the reverse to a Brownian bridge between a target distribution and a point, amounting to approximating the reverse to particular kind of diffusion process. The trajectory balance objective – which directly optimizes for agreement of the forward and reverse path measures – is equivalent in expected gradient to the objective of the path integral sampler (Zhang & Chen, 2022) when trained using on-policy forward exploration, but can also be trained using off-policy trajectories to accelerate mode discovery, which was found to be beneficial in Lahlou et al. (2023); Malkin et al. (2023).

Extending the setup of Zhang & Chen (2022); Lahlou et al. (2023), we consider the problem of sampling from a mixture of 25 well-separated Gaussians (see Figure 5), with horizontal and vertical spacing of 5 between the component means and each component having variance 0.3. The learned sampler integrates the SDE $d\mathbf{x}_t = \boldsymbol{\mu}(\mathbf{x}, t) \, dt + 5 \, d\mathbf{w}_t$, where $\boldsymbol{\mu}$ is the output of a neural network (a small MLP) taking $\mathbf{x}$ and $t$ as input, with initial condition $\mathbf{x}_0 = \mathbf{0}$ up to time $t = 1$. The reward for $\mathbf{x}_1$ is the density of the target distribution. The neural network architecture and training hyperparameters are the same as in Lahlou et al. (2023).

---

[7]We use the publicly available implementations of these reward functions at this URL.

We generate a dataset $\mathcal{D}$ of 2715 approximate samples from the target distribution first using GGNS, and then we use bootstrapping to generate equally weighted samples, using the bootstrapping algorithm in Handley (2019). We consider five algorithms for training the SDE drift $\mu$:

(1) **On-policy TB:** We train $\mu$ by optimizing the trajectory balance objective on trajectories obtained by integration of the SDE being trained (equivalent to the path integral sampler objective and to minimization of the KL divergence between forward and reverse path measures).

(2) **Exploratory TB:** We optimize the trajectory balance objective on trajectories obtained from a noised version of the SDE, which adds Gaussian noise with standard deviation $\sigma$ to the drift term at each step. Consistent with Lahlou et al. (2023), we linearly reduce $\sigma$ from 0.1 to 0 over the first 2500 training iterations. Such exploration is expected to aid in discovery of modes.

(3) **Backward TB:** We optimize the trajectory balance objective on trajectories sampled from the reverse (diffusion) process begun at samples from $\mathcal{D}$.

(4) **Backward MLE:** We sample trajectories from the reverse process begun at samples from $\mathcal{D}$ and train $\mu$ so as to maximize the log-likelihood of these trajectories under the forward process. This objective amounts to training a diffusion model or score-based generative model (Song & Ermon, 2019; Ho et al., 2020) on $\mathcal{D}$, as the optimal $\mu$ is the score of the target distribution convolved with a Gaussian and appropriately scaled.

(5) **Forward + backward TB:** We optimize the trajectory balance objective both on trajectories obtained by integrating the SDE forward from samples from $\mathcal{D}$ and on reverse trajectories begun at samples from $\mathcal{D}$. This method resembles the training policy used by Zhang et al. (2022).

KDE plots of samples from the trained models, as well as training metric plots, are shown in Figure 5. Training with on-policy TB alone (1) results in mode collapse, a typical effect of training with a reverse KL objective (Malkin et al., 2023). We see that while noise introduced in forward exploration (2) helps mode discovery, it is insufficient for all modes to be found. Training using trajectory balance on backward trajectories (3) results in spurious modes, as the model is unlikely to see states that are far from those seen along reverse trajectories from $\mathcal{D}$ during training. Maximum (4) discovers all modes of the distribution, as they are represented in $\mathcal{D}$, but closer inspection reveals that they are not modeled as accurately; this effect is more pronounced when the dataset $\mathcal{D}$ is small. The best sampling performance is reached by models that perform a mix of forward exploration and reverse trajectories from the dataset samples.

It is important to note that with well-tuned exploratory policies – as in (2) – it is possible to coax the model into discovering all of the modes and modeling them with high fidelity. However, the model is highly sensitive to the exploration parameters: if the exploration rate is too high or not reduced slowly enough, the model is slow to converge and blurs of 'fattens' the modes, while an exploration rate that is too low results in mode collapse. On the other hand, mixing forward exploration with backward trajectories from the approximate samples allows the sampler to model all of the modes accurately without such tuning. Notably, we found that the forward trajectories in (5) can be sampled either on-policy or from a tempered policy, with little difference in performance.

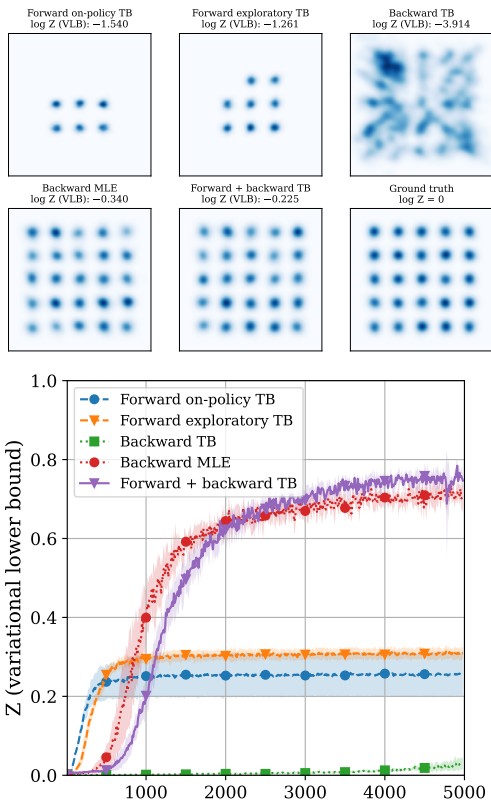

Figure 5: **Above:** KDE plots of samples from trained stochastic control models (ground truth distribution at lower right). Mixing forward sampling with noising trajectories initialized at nested sampling outputs results in all modes being modelled accurately. **Below:** Variational lower bound on the partition function for the five samplers. The theoretical maximum – achieved by the Schrödinger bridge between the Dirac distribution at the origin and the target distribution – is $Z = 1$.

## 6 DISCUSSION AND CONCLUSIONS

We have introduced a new nested sampling algorithm based on Hamiltonian Slice Sampling. Gradient-guided nested sampling improves on previous nested sampling algorithms by removing the linear dependence of the number of live points on dimensionality. It also makes use of the power of differentiable programming frameworks and parallelization for significant speed improvements. We have shown that the proposed method scales much better with dimensionality than other nested sampling algorithms, thanks to the use of gradient information. This better scaling allows us to apply nested sampling in high-dimensional problems that were too computationally expensive for previous methods. We have also shown that GGNS can be combined with generative flow networks to obtain large numbers of samples from complex posterior distributions. Applications of GGNS to difficult real-world inference problems, both on its own and in combination with amortized sampling methods, are left for future work.

## REPRODUCIBILITY STATEMENT

We include with this submission an implementation of GGNS in PyTorch (Paszke et al., 2019), along with notebooks to reproduce the results from the experiments, in the supplementary materials.

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

---

**Algorithm 1** A simple nested sampling algorithm. Note that more sophisticated implementations include clustering of live points, a calculation of the error in the estimate of $\log Z$, and other techniques to improve performance.

---

1: Initialise $n_{\text{live}}$ live points from the prior $\pi(\theta)$.
2: Initialise an empty set of dead points.
3: Evaluate the likelihood $\mathcal{L}_i = \mathcal{L}(\theta_i)$ for each live point.
4: Set $X = 1$.
5: Set $Z = 0$.
6: **while** $X_N > \text{tol} \cdot Z$ **do**
7:    Select the live point with the lowest likelihood $\mathcal{L}_j$, and move it from the set of live points to the set of dead points.
8:    Sample a new point $\theta_{\text{new}}$ from the prior $\pi(\theta)$, under the condition $\mathcal{L}(\theta_{\text{new}}) > \mathcal{L}_j$.
9:    Set $Z \to Z + \frac{1}{n_{\text{live}}+1} X \mathcal{L}_j$.
10:    Set $X \to X \frac{n_{\text{live}}}{n_{\text{live}}+1}$.
11:    Add $\theta_{\text{new}}$ to the set of live points.
12: **end while**
13: **for** i = 1, ..., $n_{\text{live}}$ **do**
14:    Select the live point with the lowest likelihood $\mathcal{L}_j$, and move it from the set of live points to the set of dead points.
15:    Set $Z \to Z + \frac{1}{n_{\text{live}}+1} X \mathcal{L}_j$.
16:    Set $X \to X \frac{n_{\text{live}}}{n_{\text{live}}+1}$.
17: **end for**

---

## A    NESTED SAMPLING REVIEW

Nested sampling was initially introduced as a method to calculate the Bayesian evidence or marginal likelihood:

$$\mathcal{Z} = \int \mathcal{L}(\theta)\pi(\theta)d\theta, \tag{7}$$

where $\mathcal{L}(\theta)$ is the likelihood function, and $\pi(\theta)$ is the prior distribution.

The key idea of nested sampling is to define a new variable called the *cumulative prior mass* or the *prior volume* as:

$$X(\theta) = \int_{\mathcal{L}(\theta') > \mathcal{L}(\theta)} \pi(\theta')d\theta', \tag{8}$$

which is the fraction of the prior mass that has a likelihood greater than the likelihood of the current point. This variable is bounded between 0 and 1, and can be used to rewrite the evidence as:

$$\mathcal{Z} = \int_0^1 \mathcal{L}(X)dX, \tag{9}$$

which is a one-dimensional integral. Therefore, we can evaluate the likelihoods of a set of points $\{\theta_i\}$ sorted by their likelihood, and use them to estimate the evidence by approximating the integral in eq. (9) as a sum:

$$\mathcal{Z} \approx \sum_i \mathcal{L}(X_i)\Delta X_i, \tag{10}$$

where $\Delta X_i = X_{i-1} - X_i$ is the difference in prior volume between the $i$-th and the $(i-1)$-th point. This approximation is exact in the limit of an infinite number of points, and can be used to estimate the evidence to arbitrary precision.

The key idea of nested sampling is the following: We start by sampling a set of $n_{\text{live}}$ *live points* from the prior distribution. We then find the point with the lowest likelihood, and remove it from the set, adding it to the set of *dead points*. We then replace it with a new point sampled from the prior, subject to the constraint that its likelihood is greater than the likelihood of the point that

was removed. This means that, while it is unfeasible to calculate $X(\theta)$ for each of the new points exactly, we can approximate it by using the fact that, at each iteration, the prior volume is contracted by approximately:

$$\Delta X_i \approx \frac{n_{\text{live}}}{n_{\text{live}} + 1}.$$

(11)

This process is repeated until the remaining posterior mass is smaller than some fraction of the current estimate of $\mathcal{Z}$. The set of points that we have sampled can then be used to estimate the evidence using eq. (10).

Furthermore, GGNS can be used for parameter inference. To do that, we assign the following importance weight to each point:

$$p_i = \frac{\mathcal{L}(w_i)}{Z},$$

(12)

where $w_i$ is the prior volume of the shell that was used to sample the $i$-th point:

$$w_i = X_{i-1} - X_i.$$

(13)

An example implementation of a nested sampling algorithm is shown in algorithm 1.

## B  HAMILTONIAN SLICE SAMPLING ALGORITHM

---

**Algorithm 2** Hamiltonian or Reflective Slice Sampling

---
1: Choose a point $\theta$ from the existing set of live points.
2: Choose a direction $d$.
3: Choose an initial momentum $p_{\text{ini}} \sim \mathcal{N}(0, 1)$.
4: Set $p = p_{\text{ini}}$.
5: Set $x = \theta$.
6: Set $t = 0$.
7: **while** $t < T$ **do**
8:     Set $x = x + p \, dt$.
9:     **if** $x$ is outside the slice **then**
10:         Take $n = \nabla\mathcal{L}(\theta)/\|\nabla\mathcal{L}(\theta)\|$
11:         Set $p = p - 2(p \cdot n)n$.
12:     **end if**
13:     Set $t = t + dt$.
14: **end while**
15: Set $\theta' = x$.

---

We show an example implementation of the Hamiltonian or Reflective Slice Sampling algorithm in algorithm 2.

## C  HYPERPARAMETERS OF GGNS

Table table 2, shows the different hyperparameters of GGNS. This shows the little tuning required for GGNS to perform unbiased sampling.

## D  COMPARISON WITH FLOW ANNEALED IMPORTANCE SAMPLING BOOSTRAP

We show the comparison with Flow Annealed Importance Sampling Boostrap (FAB) on the mixture of 40 Gaussians used in Midgley et al. (2022), in Figure 6. The image shows GGNS samples all the modes of the distribution more accurately than FAB. We also show in appendix D the results from GGNS in the torus reward introduced in eq. (6). We see that GGNS can successfully sample the distribution. We do not show a comparison with FAB on this task, as we could not easily train it on a torus.

Table 2: Log-evidence function estimation bias (mean and standard deviation over 10 runs). The first rows are from our method, while the rest are from Zhang & Chen (2022); Lahlou et al. (2023). Note that the last three methods are using importance-weighted bound $B_{RW}$. In bold font, we show the estimates that are unbiased at the one standard deviation level.

| Parameter | Default Value | Description |
|---|---|---|
| n_live | Number of live points. A higher number leads to better mode coverage. | 200[8] |
| tol | Tolerance. The stopping criterion. GGNS terminates when $\mathcal{L}_i X_i / \max(\mathcal{L}_i X_i) <$ tol. | 0.01 |
| min_ref | The minimum number of reflections. We sample points after they have reflected of the boundary at least min_ref. | 1 |
| max_ref | The maximum number of reflections. We stop each HSS iteration after the point has reflected max_ref times off the boundary. | 3 |
| delta_p | The number of noise added to the momentum at each HSS step, to decorrelate samples faster. | 0.05 |

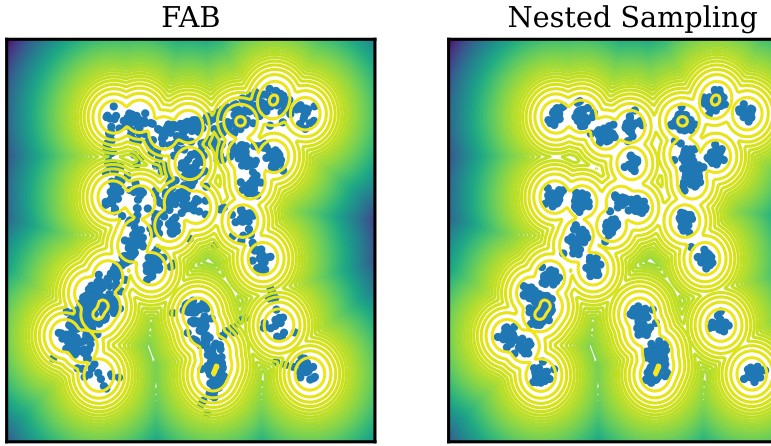

Figure 6: Contour lines for the target distribution, and samples, for the mixture of 40 Gaussians from Midgley et al. (2022). On the left, we show results for Flow Annealed Importance Sampling Boostrap (FAB), and on the right for GGNS.

# E   SAMPLING COMPLEX DISTRIBUTIONS

We further test the capacity of GGNS to model several complex distributions that are for different reasons challenging for inference algorithms. For these examples, we use a visual comparison with samples from the true distribution. We increase the number of live points to 2000 for these examples, to ensure that we have enough samples to compare with the true distribution. Because nested sampling produces weighted samples, all the figures use an alpha blending of the samples, with the alpha value proportional to the importance weight.

Firstly, we re-use the Gaussian mixture distribution from the previous example, but we increase the number of modes to 81. This distribution is difficult due to its very high multimodality. The results are shown in the top panel of Figure 8, where we see that our method successfully recovers all modes.

Secondly, the "five Swiss rolls" example consists of five copies of the "Swiss roll" distribution in two dimensions. It combines multimodality with the difficulty of sampling the complex structure

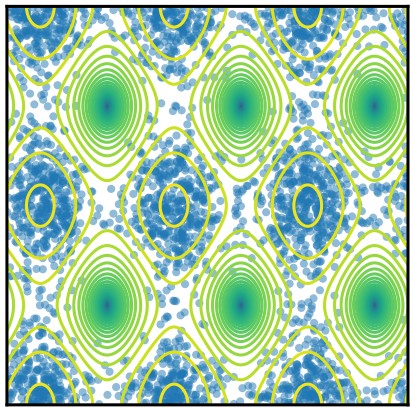

Figure 7: Contour lines for the target distribution, and `GGNS` samples, for the torus reward eq. (6). We do not show Flow Annealed Importance Sampling Boostrap (FAB) samples for this task, as we failed to train it successfully.

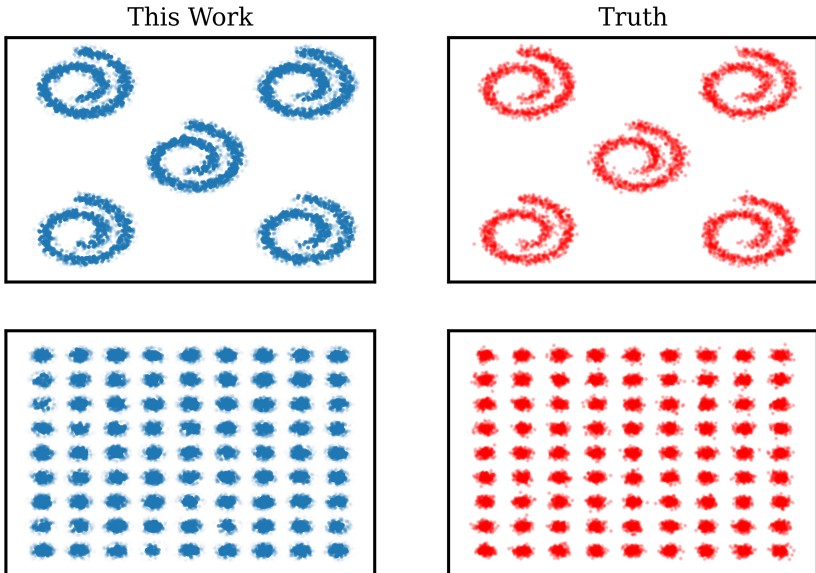

Figure 8: Comparison between the proposed method (left) and the truth (right) on the "five Swiss rolls" distribution. We show our method successfully recovers every model of this highly multi-modal distribution.

of each mode of the distribution. As shown in the bottom panel of Figure 8, the proposed method successfully samples the distribution.

## F  TORUS REWARD FUNCTION

### F.1  NORMALIZATION

As stated in the main text, for the following reward:

$$R_n(\mathbf{x}, \alpha, \beta, c) = \left( \sum_{i \text{ even}}^{n} \sin(\alpha x_i) + \sum_{j \text{ odd}}^{n} \cos(\beta x_j) + c \right)^3, \qquad x_i \in [0, 2\pi) \tag{14}$$

the normalization, defined as:

$$Z_n(\alpha, \beta, c) = \int R_n(\alpha, \beta, c) \, d\mathbf{x} \tag{15}$$

This can be found with the following recursive formula:

$$Z_n(\alpha, \beta, c) = 2\pi Z_{n-1}(\alpha, \beta, c) + 3\pi c (2\pi)^{n-1}, \qquad Z_1(\alpha, \beta, c) = 2\pi c^3 + 3\pi c. \tag{16}$$

As long as $\alpha, \beta \in \mathbb{Q}$

The easiest way to prove it is by induction: It is trivially true for $Z_1$:

$$\int (\sin(\alpha x) + c)^3 \, dx = 2\pi c^3 + 3\pi c, \qquad \forall \alpha \in \mathbb{Q}, \, \forall c. \tag{17}$$

Note that this also applies if we swap a sine for a cosine, which will become relevant later.

So, all that is left is to prove that, if the statement holds for $Z_{n-1}$, it also holds for $Z_n$. Lets assume, without loss of generality, that $n$ is even

$$Z_n(\alpha, \beta, c) = \int \left( \sum_{i \text{ even}}^{n} \sin(\alpha x_i) + \sum_{j \text{ odd}}^{n} \cos(\beta x_j) + c \right)^3 \, dx_1...dx_n \tag{18}$$

$$= \int \left[ \int \left( \sin(a x_n) + \sum_{i \text{ even}}^{n-1} \sin(\alpha x_i) + \sum_{j \text{ odd}}^{n-1} \cos(\beta x_j) + c \right)^3 \, dx_n \right] dx_1...dx_{n-1}. \tag{19}$$

Let's define:

$$K \equiv \sum_{i \text{ even}}^{n-1} \sin(\alpha x_i) + \sum_{j \text{ odd}}^{n-1} \cos(\beta x_j) + c. \tag{20}$$

Note that $K$ does not depend on $x_n$. With this:

$$Z_n(\alpha, \beta, c) = \int \left[ \int (\sin(a x_n) + K)^3 \, dx_n \right] dx_1...dx_{n-1}. \tag{21}$$

The key realization here is that the thing inside the bracket is the same as eq. (17). Therefore:

$$Z_n(\alpha, \beta, c) = \int \left( 2\pi K^3 + 3\pi K \right) dx_1...dx_{n-1}. \tag{22}$$

$$= 2\pi \int K^3 dx_1...dx_{n-1} + 3\pi \int K dx_1...dx_{n-1} \tag{23}$$

The first integral is simply $Z_{n-1}(\alpha, \beta, c)$. The second is

$$\int K dx_1...dx_{n-1} = \int \left[ \sum_{i \text{ even}}^{n-1} \sin(\alpha x_i) + \sum_{j \text{ odd}}^{n-1} \cos(\beta x_j) + c \right] dx_1...dx_{n-1} \tag{24}$$

$$= (2\pi)^{n-1} c, \tag{25}$$

because all the sine and cosine integrals cancel. Therefore:

Figure 9: Change in the Kullback-Leibler divergence with dimensionality, for the molecule example shown in Section 4.4. As described by eq. (3), the error in the estimate of $\log Z$ is proportional to $\mathcal{D}_{KL}(\mathcal{P}|\Pi)$, therefore, the unusual decrease of $\mathcal{D}_{KL}(\mathcal{P}|\Pi)$ with dimensionality, explains the reduced errors on high dimensions for the estimate of $\log Z$.

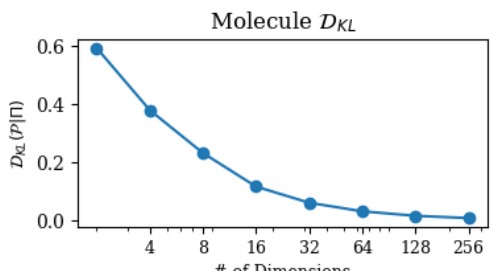

$$Z_n(\alpha, \beta, c) = 2\pi \int K^3 \mathrm{d}x_1...\mathrm{d}x_{n-1} + 3\pi \int K \mathrm{d}x_1...\mathrm{d}x_{n-1} \tag{26}$$

$$= 2\pi Z_{n-1}(\alpha, \beta, c) + 3\pi c (2\pi)^{n-1} \tag{27}$$

Q.E.D.

### F.2 BEHAVIOUR ON DIFFERENT DIMENSIONS

Our nested sampling $\log Z$ estimates for the synthetic molecule reward, show an unusual pattern of behaviour, as we can see in Section 4.4: The error goes down as the dimensionality increases. This section offers some intuition about why this happens. As described by eq. (3), the nested sampling error in the estimate of $\log Z$ is proportional to $\sqrt{\mathcal{D}_{KL}(\mathcal{P}|\Pi)/n_{\text{live}}}$. Because our problem keeps $n_{\text{live}}$ fixed, observed behaviour is due to a change in $\mathcal{D}_{KL}(\mathcal{P}|\Pi)$ as the dimensionality increases.

$\mathcal{D}_{KL}(\mathcal{P}|\Pi)$ is a measure of how much information we gain when we go from the prior to the posterior. Therefore, in more cases, it increases with dimensionality. However, as shown by appendix E, in this particular example, $\mathcal{D}_{KL}(\mathcal{P}|\Pi)$ goes down with dimensionality, which explains why the error goes down with dimensionality in Section 4.4. The reason why $\mathcal{D}_{KL}(\mathcal{P}|\Pi)$ is likely caused by cancellations in the various terms of the sum eq. (6), as dimensionality increases, but will be further explored in future work.

## G ABLATION STUDY

Given the multiple different components included in our algorithm, described in Section 3, it is important to perform an ablation study to fully understand how the different parts of the algorithm contribute to the observed improvements in performance. Therefore, this section removes each of the improvements introduced in Section 3 one by one and analyses the effect of these changes on the sampling performance. We use the task introduced in Section 4.1 to perform this comparison unless otherwise specified.

### G.1 ADAPTIVE TIME STEP CONTROL

We repeat the analysis of Section 4.1 with a fixed time step in the Hamiltonian slice sampling steps instead of using adaptive time step control. We attempt three different time steps: $\mathrm{d}t = 0.5$ and $\mathrm{d}t = 0.1$. Note that, in Section 4.1, we start with $\mathrm{d}t = 0.1$ but adapt it as the algorithm progresses.

The results, shown in Figure 10, show that when we use a large $\mathrm{d}t$ we can reduce the number of likelihood evaluations, as we achieve the minimum number of reflections faster, but we get a biased estimate of $\log Z$, as we fail to appropriately sample each slice. For small $\mathrm{d}t$, on the other hand, we get less bias in $\log Z$, but the number of likelihood evaluations goes up.

In general, the advantage of the adaptive step is that as the algorithm progresses, the volume of the region defined by eq. (4) we are exploring decreases. Therefore, a step size that is appropriate at a given point will become too large eventually as the algorithm progresses.

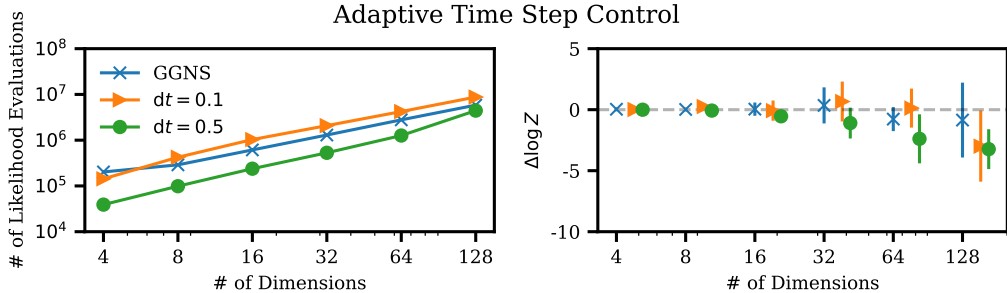

Figure 10: Comparison of likelihood evaluations (left panel) and error in the estimation of $\log \mathcal{Z}$ for different dimensionalities between baseline `GGNS` (blue), and `GGNS` without adaptive step control ($dt = 0.5$ in green and $dt = 0.1$ in orange)

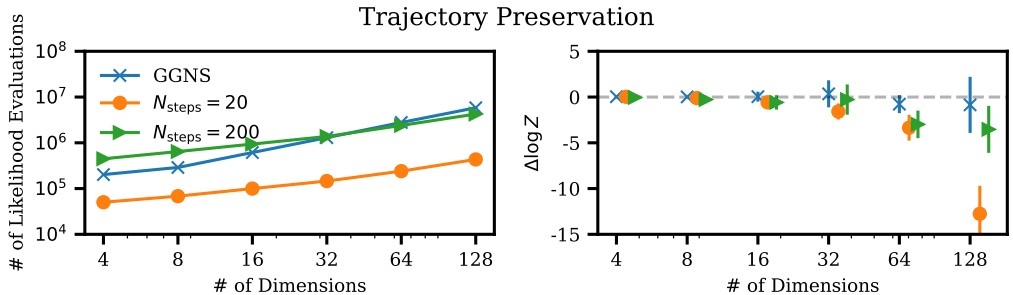

Figure 11: Comparison of likelihood evaluations (left panel) and error in the estimation of $\log \mathcal{Z}$ for different dimensionalities between baseline `GGNS` (blue), and `GGNS` without trajectory preservation ($N_{\text{steps}} = 200$ in green and $N_{\text{steps}} = 20$ in orange). Both of these lead to correlated estimates of $\log Z$, as shown by the right panel.

## G.2 TRAJECTORY PRESERVATION

`GGNS` uses a novel approach to sample the trajectories and to ensure that samples are correlated, where we ensure a certain number of boundary reflections. We compare what happens when we use the simpler approach of integrating our trajectory for a fixed number of steps $n_{\text{steps}}$ and simply keeping the last sample. We repeat the analysis for $n_{\text{steps}} = 20$ and $n_{\text{steps}} = 200$.

Figure 11 shows the results. We see that a fixed number of steps leads to a biased estimate of $\log Z$. The argument for this, similarly to what it was for trajectory preservation, is that the volume of the region eq. (4) decreases as the algorithm progresses.

We also study what happens when we use trajectory preservation but do not add noise $\delta_{\text{p}}$ to achieve a faster decorrelation of the samples. The results, shown in Figure 12, are intuitive: No noise in the trajectories reduces the likelihood evaluations, as trajectories are less noisy but lead to biased evidence estimates, as the samples are not fully decorrelated.

## G.3 MODE COLLAPSE MITIGATION

To study the effect of this, we need a multimodal distribution. We use a mixture of nine Gaussians, shown in Figure 13. To study the effect of our mode collapse mitigation, we run nested sampling on this problem, with and without this setting. The main hyperparameter that affects the number of modes found, is the number of live points $n_{\text{live}}$. Therefore, we run nested sampling on this problem for different values of this hyperparameter. For each configuration, we run the algorithm 10 times,

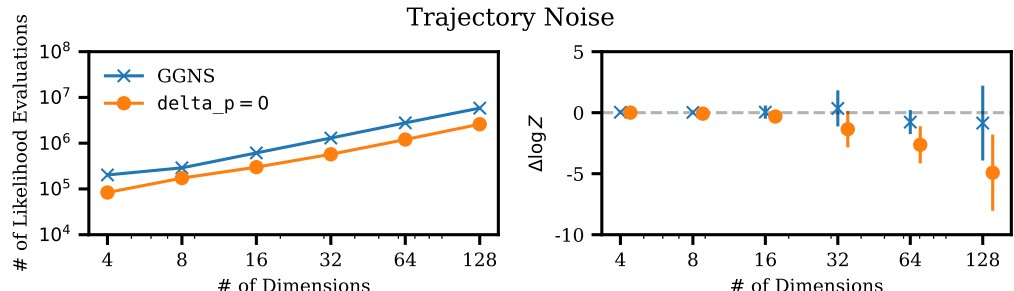

Figure 12: Comparison of likelihood evaluations (left panel) and error in the estimation of $\log \mathcal{Z}$ for different dimensionalities between baseline `GGNS` (blue), and `GGNS` without adding noise to the trajectories (orange). Less noise decreases the number of evaluations but leads to a biased $\log Z$ estimate.

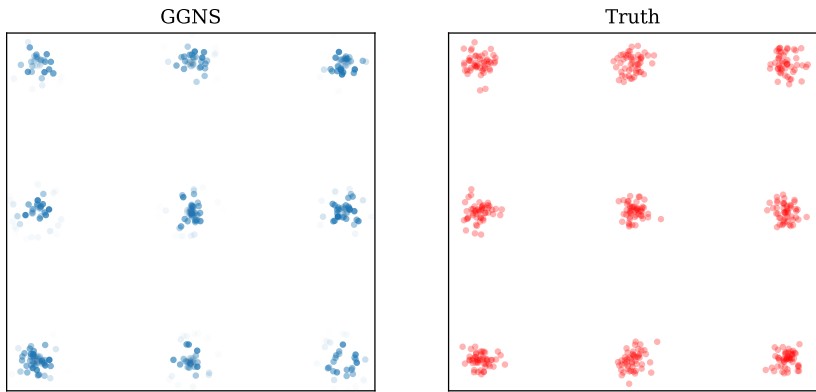

Figure 13: The distribution used for our ablation study on mode collapse mitigation, true samples on the right, and samples with GGNS ($n_{\text{live}} = 100$) on the left.

and count the number of modes found. We define a mode as being found, if at least one of the samples is within a distance $\sigma$ of the center of the mode.

The results are shown in table 3. We see how, generally, mode collapse mitigation helps us find a higher number of modes. Although the number of live points is the most important hyperparameter when it comes to mode finding, the ability to find all modes for a fixed $n_{\text{live}}$ is higher when using mode collapse mitigation.

### G.4 TERMINATION CRITERION

Finally, we repeat the analysis of Section 4.1 using the termination criterion used by other nested sampling algorithms such as `DyNesty` and `PolyChord`, in which we terminate the algortihm when $\mathcal{L}_{\text{max}} X_i < \text{tol}$, for some tolerance hyperparameter. We used a tolerance 0.01, a value often used by nested sampling practitioners.

We show the results in Figure 14. The number of likelihood evaluations appears similar, but at high dimensions, the previous termination leads to a biased estimate of $\log Z$. Indeed, while the number of likelihood estimations is of the same order of magnitude, the termination used by `GGNS` leads to slightly more evaluations ($\sim 5.8 \cdot 10^6$ )than the previous one ($\sim 5.4 \cdot 10^6$ ) for $d = 128$. These $400, 000$ evaluations are likely to drive the underestimation of the evidence by the previous method.

Table 3: The average number of modes found over 10 nested sampling runs, sampling the distribution shown in Figure 13, with and without mode collapse mitigation, for varying number of live points.

| Method | $n_{\text{live}} = 20$ | $n_{\text{live}} = 50$ | $n_{\text{live}} = 100$ | $n_{\text{live}} = 200$ |
|---|---|---|---|---|
| Without Mode Collapse Mitigation | 3.6 | 6.4 | 8.2 | 8.9 |
| With Mode Collapse Mitigation | 4.1 | 6.4 | 8.4 | 9 |

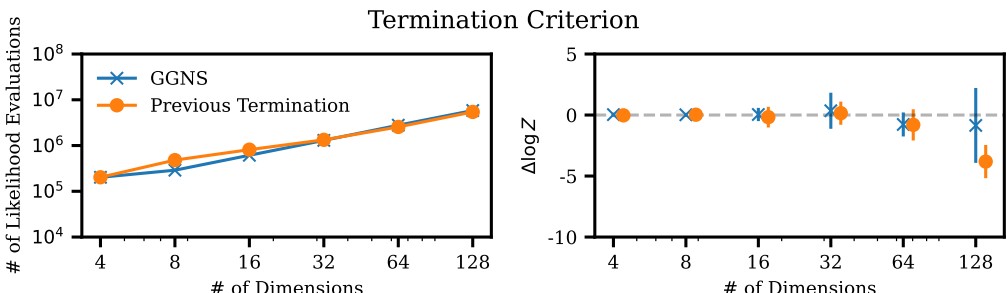

Figure 14: Comparison of likelihood evaluations (left panel) and error in the estimation of $\log \mathcal{Z}$ for different dimensionalities between baseline `GGNS` (blue), and `GGNS` using the termination criterion used by `MultiNest`, `PolyChord` and `DyNesty`.

### G.5 OTHER CHANGES

We could not study other changes, such as the pruning mechanism or parallel Evolution of live points, as this would have led to a full rewrite of the algorithm. We leave this study for future work.

### G.6 CONCLUSIONS

The main conclusion of this ablation study is that naive changes to `GGNS` quickly lead to biased sampling. It is the combination of the contributions introduced in Section 3 that leads to robust evidence on high dimensions. Of course, it is true that for each of these parts, there are settings that will work for any problem, i.e. we can always make the step size small enough, the number of steps high enough, etc. However, the main advantage of our algorithm is that it works without the need for fine-tuning all these parameters.

## H  GGNS ALGORITHM

We show the full `GGNS` algorithm in algorithm 3. Our cluster-finding algorithm and our Hamiltonian slice sampling algorithm are shown in algorithm 4 and algorithm 5 respectively and are both used in algorithm 3. The cluster statistics formalism presented in this section follows (Handley et al., 2015b). We refer the reader to the original paper for derivations on where these formulas come from.

The summary statistics are initiated using the following equation:

---

**Algorithm 3** The `GGNS` algorithm

---

1: Initialise $n_{\text{live}}$ live points from the prior $\pi(\theta)$.
2: Initialise an empty set of dead points.
3: Evaluate the likelihood $\mathcal{L}_i = \mathcal{L}(\theta_i)$ for each live point.
4: Initiate summary statistics $\left\{\overline{Z}, \overline{X}, \overline{Z_p}, ...\right\}$, using eq. (28) to eq. (39).
5: Set $\Delta(X\mathcal{L}) = 0$.
6: Set $(X\mathcal{L})_{\text{max}} = 0$.
7: Set $n_{\text{clusters}} = 1$.
8: Set $dt = dt_{\text{ini}}$.
9: **while** $\Delta(X\mathcal{L}) > $ tol **do**
10:     **for** i = 1, ..., $n_{\text{clusters}}$ **do**
11:         Use cluster finding algorithm 4
12:         If new clusters are found, initiate them by splitting the cluster $i$, using eq. (50) to eq. (56)
13:     **end for**
14:     **for** j = 1, ..., $n_{\text{live}}//2$ **do**
15:         Select the point with the lowest likelihood $\mathcal{L}_j$ and remove them from the set of live points to the set of dead points.
16:         Update the summary statistics, using eq. (40) to eq. (49).
17:         If a cluster has no points, remove it, and set $n_{\text{clusters}} - = 1$
18:     **end for**
19:     Generate the cluster labels for the next points $n_{\text{live}}//2$, proportionally to $\overline{X_p}$
20:     Sample $x \sim \theta_{\text{live}}$, from the appropriate clusters
21:     Use algorithm 5, to get $\theta_{\text{new}}$, and `out_frac`, under the condition $\mathcal{L}(\theta_{\text{new}}) > \mathcal{L}_j \; \forall \theta_{\text{new}}$.
22:     **if** `out_frac` $> 0.15$ **then**
23:         Set $dt = dt * 0.9$
24:     **else if** `out_frac` $< 0.05$ **then**
25:         Set $dt = dt * 1.1$
26:     **end if**
27:     Add $\theta_{\text{new}}$ to the set of live points.
28:     Set $(X\mathcal{L})_{\text{max}} = \max\left(X\mathcal{L}_{\text{max}}, (X\mathcal{L})_{\text{max}}\right)$, where $\mathcal{L}_{\text{max}}$ is the maximum likelihood amongst the live points.
29:     Set $(\Delta X\mathcal{L}) = X\mathcal{L}_{\text{max}}/(X\mathcal{L})_{\text{max}}$
30: **end while**
31: **for** i = 1, ..., $n_{\text{live}}$ **do**
32:     Select the live point with the lowest likelihood $\mathcal{L}_j$ and move it from the set of live points to the set of dead points.
33:     Set $Z = Z + \frac{1}{n_{\text{live}}+1} X\mathcal{L}_j$.
34:     Set $X = X \frac{n_{\text{live}}}{n_{\text{live}}+1}$.
35: **end for**

---

**Algorithm 4** The cluster finding algorithm used in algorithm 3, for a cluster containing $n_{points}$ points.

---

1: Initialise `prev_sizes = None`.
2: **for** k = 2, ... , $n_{points}$ **do**
3:     Run k-nearest-neighbours (KNN) on the cluster points, with value $k$
4:     Set `cluster_sizes` as the number of points in each KNN cluster
5:     **if** `cluster_sizes = prev_sizes` **then**
6:         Break
7:     **else**
8:         Set `prev_sizes = cluster_sizes`
9:     **end if**
10: **end for**
11: **return** The number of KNN clusters.

---

---

**Algorithm 5** The Hamiltonian slice sampling algorithm used in algorithm 3, starting from $n$ points with position $x$, and with step size $dt$; and with a likelihood barrier $\mathcal{L}_{\min}$

---

1: Set `num_out_steps` $= 0$, `num_in_steps` $= 0$
2: Set $p \sim \mathcal{N}(0, 1)$.
3: Set `num_reflections`$[1, ..., n] < -0$
4: Set `x_saved` $= \{\}$
5: **while** $\min($`num_reflections`$) <$ `max_reflections` **do**
6:     Set $x+ = p * dt$
7:     Call the likelihood function, to get $\mathcal{L}$ and $\nabla\mathcal{L}$
8:     Set `outside`$[1, ..., n] = \mathcal{L} < \mathcal{L}_{\min}$
9:     Take $n = \nabla\mathcal{L}/\|\nabla\mathcal{L})\|$
10:     Set $p[$`outside`$] = p[$`outside`$] - 2(p \cdot n)n[$`outside`$]$.
11:     Set $\epsilon \sim \mathcal{N}(0, 1)$
12:     Set $p = p * (1 + \epsilon *$ `delta_p`$)$
13:     Set `num_reflections`$+ =$ `outside`
14:     **if** $\min($`num_reflections`$) <$ `min_reflections` **then**
15:         Add $x[\sim$ `outside`$]$ to `x_saved`
16:     **end if**
17:     Set `num_out_steps`$+ = \sum($`outside`$)$
18:     Set `num_in_steps`$+ = \sum(\sim$ `outside`$)$
19: **end while**
20: Set `out_frac` $=$ `num_out_steps`$/($`num_out_steps` $+$ `num_in_steps`$)$
21: Samples $\theta \sim$ `x_saved`
22: **return** $\theta$, `out_frac`

---

$$\overline{Z} = 0, \tag{28}$$

$$\overline{Z_p} = \left\{\overline{z}\right\}, \tag{29}$$

$$\overline{Z^2} = 0, \tag{30}$$

$$\overline{Z_p^2} = \left\{\overline{Z^2}\right\}, \tag{31}$$

$$\overline{ZX} = 0, \tag{32}$$

$$\overline{ZX_p} = \left\{\overline{ZX}\right\}, \tag{33}$$

$$\overline{Z_pX_p} = \left\{\overline{ZX}\right\}, \tag{34}$$

$$\overline{X} = 1, \tag{35}$$

$$\overline{X_p} = \left\{\overline{X}\right\}, \tag{36}$$

$$\overline{X_p^2} = \left\{\overline{X^2}\right\}, \tag{37}$$

$$\overline{X_pX_q} = 0 \quad (q \neq p), \tag{38}$$

$$\tag{39}$$

where $p$ and $q$ refer to the cluster numbers, initially 1. To update the summary statistics, we use:

$$\overline{\mathcal{Z}} \rightarrow \overline{\mathcal{Z}} + \frac{\overline{X}_p \mathcal{L}}{n_p + 1}, \tag{40}$$

$$\overline{\mathcal{Z}}_p \rightarrow \overline{\mathcal{Z}}_p + \frac{\overline{X}_p \mathcal{L}}{n_p + 1}, \tag{41}$$

$$\overline{X}_p \rightarrow \frac{n_p \overline{X}_p}{n_p + 1}, \tag{42}$$

$$\overline{\mathcal{Z}^2} \rightarrow \overline{\mathcal{Z}^2} + \frac{2\overline{\mathcal{Z}X_p}\mathcal{L}_p}{n_p + 1} + \frac{2\overline{X_p^2}\mathcal{L}^2}{(n_p + 1)(n_p + 2)}, \tag{43}$$

$$\overline{\mathcal{Z}_p^2} \rightarrow \overline{\mathcal{Z}_p^2} + \frac{2\overline{\mathcal{Z}_pX_p}\mathcal{L}}{n_p + 1} + \frac{2\overline{X_p^2}\mathcal{L}^2}{(n_p + 1)(n_p + 2)}, \tag{44}$$

$$\overline{\mathcal{Z}X_p} \rightarrow \frac{n_p\overline{\mathcal{Z}X_p}}{n_p + 1} + \frac{n_p\overline{X_p^2}\mathcal{L}}{(n_p + 1)(n_p + 2)}, \tag{45}$$

$$\overline{\mathcal{Z}X_q} \rightarrow \overline{\mathcal{Z}X_p} + \frac{\overline{X_pX_q}\mathcal{L}}{(n_p + 1)} \qquad (q \neq p), \tag{46}$$

$$\overline{\mathcal{Z}_pX_p} \rightarrow \frac{n_p\overline{\mathcal{Z}_pX_p}}{n_p + 1} + \frac{n_p\overline{X_p^2}\mathcal{L}}{(n_p + 1)(n_p + 2)}, \tag{47}$$

$$\overline{X_p^2} \rightarrow \frac{n_p\overline{X_p^2}}{n_p + 2}, \tag{48}$$

$$\overline{X_pX_q} \rightarrow \frac{n_p\overline{X_pX_q}}{n_p + 1} \qquad (q \neq p). \tag{49}$$

When we need to split a cluster $p$ into multiple clusters $i$, we use:

$$\overline{X}_i = \frac{n_i}{n}\overline{X}_p, \tag{50}$$

$$\overline{X_i^2} = \frac{n_i(n_i + 1)}{n(n + 1)}\overline{X_p^2}, \tag{51}$$

$$\overline{X_iX_j} = \frac{n_in_j}{n(n + 1)}\overline{X_p^2}, \tag{52}$$

$$\overline{X_iY} = \frac{n_i}{n}\overline{X_pY} \qquad Y \in \{Z, Z_p, X_q\}, \tag{53}$$

$$\overline{Z}_i = \frac{n_i}{n}\overline{Z}_p, \tag{54}$$

$$\overline{Z_iX_i} = \frac{n_i(n_i + 1)}{n(n + 1)}\overline{Z_pX_p}, \tag{55}$$

$$\overline{Z_i^2} = \frac{n_i(n_i + 1)}{n(n + 1)}\overline{Z_p^2}. \tag{56}$$

$$\tag{57}$$

## I   TERMINATION CRITERION EXPLAINED

appendix E explains the intuition behind the new termination criterion introduced in Section 3. In nested sampling, the likelihood increases as the algorithm progresses, corresponding to going from right ($X = 1$) to left ($X = 0$) in the figure. However, the product of the prior volume $X$ and the likelihood $\mathcal{L}$ always follow the same pattern: It starts low, reaches a peak, and then decreases

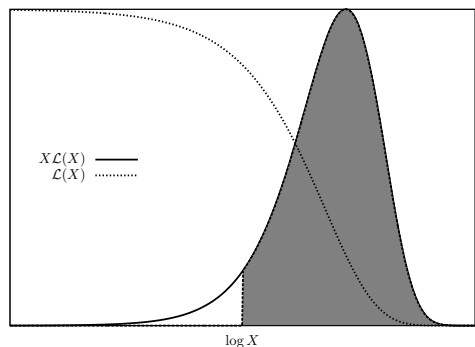

Figure 15: In nested sampling, the likelihood $\mathcal{L}$ (dotted line) goes up, as the algorithm progresses from $X = 1$ to $X = 0$ (right to left in the plot). However, the product $X\mathcal{L}(X)$ starts low, as the likelihood is small, peaks, end goes down again. Therefore, our termination criterion checks when $X\mathcal{L}(X)$ has peaked, and again gone close to zero. Image credit (Handley et al., 2015b).

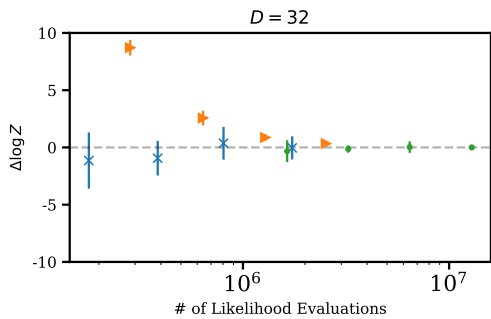
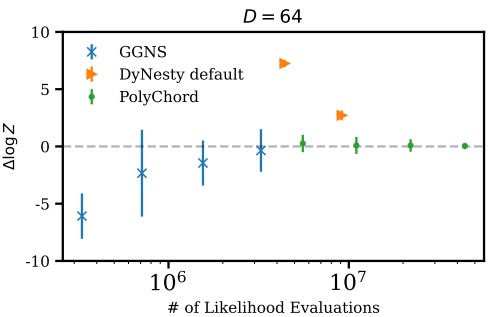

Figure 16: The bias in the estimate of $\log Z$ as a function of number of likelihood evaluations, for GGNS (blue), and other nested sampling algorithms (PolyChord in green and dynesty in orange). We achieve different numbers of like evaluations by changing the number of live points. We run each algorithm with $n_{\text{live}} = \{50, 100, 200, 400\}$. Note that the dynesty runs with $n_{\text{live}} = 50$ and $n_{\text{live}} = 100$ are not in the plot, as they are too far up in the y-axis.

towards zero again. Therefore, our termination criterion consists of checking the ratio of the current value of $X\mathcal{L}$ to the maximum value that $X\mathcal{L}$ has reached throughout the algorithm. When that fraction is smaller than some threshold, we stop the algorithm.

## J  EVIDENCE ESTIMATION AS A FUNCTION OF LIKELIHOOD EVALUATIONS

In this appendix, we study the relationship between the number of likelihood evaluations, and the estimate of $\Delta \log Z$. The easiest way to vary the number of likelihood evaluations, is by varying the number of live points used. We repeat the analysis of Section 4.1, for two values of the number of dimensions $d = 32$ and $d = 64$, for each of the algorithms; varying the number of live points in the range $n_{\text{live}} = \{50, 100, 200, 400\}$.

We see how, in both cases, dynesty can lead to very biased inference, if the number of live points is low. On the other hand, PolyChord reliably achieves unbiased inference, at the expense of a much higher number of likelihood evaluations. GGNS gets the best of each, by achieving unbiased inference with less likelihood evaluations. We also see how our algorithm scales better with dimensionality, when we compare its performance between the left and the right plots, with the other algorithms.

