# OpenReview forum: "Improving Gradient-guided Nested Sampling for Posterior Inference"
_ICLR.cc/2024/Conference — Submitted to ICLR 2024_

### Official Review · Reviewer_C7my · 2023-10-29

**Soundness:** 3 good
**Presentation:** 3 good
**Contribution:** 3 good
**Rating:** 6
**Confidence:** 3

**Summary:**

The paper proposes GGNS, a performant, general-purpose gradient-guided nested sampling algorithm that scales well with dimensionality and demonstrates competitively on a range of synthetic and real-world problems. In particular, the gradients calculated with differentiable programming are combined with HSS to propose new points, dynamic nested sampling is used for parallelization, a new termination criterion and cluster identification are also proposed. Furthermore, the authors show the potential of combining nested sampling with generative flow networks, leading to faster mode discovery and convergence of evidence estimates compared with GFlowNets.

**Strengths:**

GGNS enables the use of gradient information through the differentiable programming frameworks.
Empirically, GGNS shows the best linear scaling and performs evidence estimation accurately even as the dimensionality approaches the number of live points.
The proposed method adds practical value in the nested sampling community.
The combination of GGNS with GFlowNets opens the door to a number of interesting future research.

**Weaknesses:**

Novelty is limited to a combination of existing methods.
The proposal of using gradients in guiding the choice of new live points can be elaborated more in the Contribution section.
Experiments are almost all synthetic, and some results do not seem to be better than GFlowNets.

**Questions:**

In Figure 1, instead of plotting the error in the estimate of log Z versus the number of dimensions, how does the error grow versus the number of likelihood evaluations?

In Figure 1, the error bar (i.e., standard deviation across 10 runs) are much bigger for GGNS compared with the baselines, does it mean it is trading off variance for bias?

Table 1, all methods except GGNS are under-biased for the Gaussian mixture example, is it just by chance?

In the 'Many Wells' experiment (Figure 4), are you comparing against FAB or FAB with buffer (which should give better results than FAB)?

Minor:
Formatting under Figure 1 needs to be modified.
page 7, "due to the high its high dimensionality" -> "due to its high dimensionality"

---

> ### Author Response · Authors · 2023-11-20
> **Response to Reviewer C7my**
>
> We thank the reviewer for their very helpful feedback.
>
>
> We have tried to highlight more clearly the original contributions from this paper: Firstly, we combine existing ideas that have never been combined before, resulting in a performance in sampling tasks that beats all existing state-of-the-art methods. Secondly, we use Nested Sampling with GFlowNets to generate a large number of samples from the target distribution. As the reviewer highlighted, we believe that this opens the door to many interesting research directions.
>
> To clarify the value of the individual contributions, we have added ablation studies (Appendix G) to elucidate the value of each component. We have also highlighted more clearly how we use the gradient in guiding the choice of new live points in Section 3. We have also added an intuitive argument about why this update only requires $O(1)$ reflections, as opposed to baseline slice sampling.
>
> Regarding your questions:
>
> > In Figure 1, instead of plotting the error in the estimate of log Z versus the number of dimensions, how does the error grow versus the number of likelihood evaluations?
>
> We have added a plot of log Z vs. the number of likelihood evaluations in a separate appendix (Appendix J). GGNS needs fewer likelihood evaluations to reach the same error.
>
>
> > In Figure 1, the error bar (i.e., standard deviation across 10 runs) are much bigger for GGNS compared with the baselines, does it mean it is trading off variance for bias?
>
> Interesting point. This is true inasmuch as the other methods do seem to have less variance while being more biased. At the same time, this is not an instance of the typical bias/variance tradeoff in probabilistic modeling: the other methods are biased because they are not correctly sampling the space (e.g., by missing modes, leading to an easier sampling problem). Therefore, it is not a matter of **more** model complexity or less regularization leading to less variance, but a matter of the **simpler** sampling methods being inaccurate.
>
> > Table 1, all methods except GGNS are under-biased for the Gaussian mixture example, is it just by chance?
>
> As far as we understand, the reason other methods underestimate the normalization constant for the Gaussian mixture is that they are likely to have missed, or in some way underestimated, a mode. This leads to these methods thinking that there is less total posterior mass and, therefore, that logZ is smaller. We have added a comment about this in the table caption.
>
>
> > In the 'Many Wells' experiment (Figure 4), are you comparing against FAB or FAB with buffer (which should give better results than FAB)?
>
>
> This is FAB with buffer, we have now clarified this in the text, thank you for pointing this out.
>
> > Minor: Formatting under Figure 1 needs to be modified. page 7, "due to the high its high dimensionality" -> "due to its high dimensionality"
>
> Thank you; these have been fixed.

---

> > ### Comment · Reviewer_C7my · 2023-11-22
> >
> > Thank you for addressing my concerns and the additional clarification in the paper, I appreciate the authors' effort in the additional ablation studies which has improved the paper. I maintain a positive score.

---

### Official Review · Reviewer_vEK5 · 2023-10-31

**Soundness:** 3 good
**Presentation:** 3 good
**Contribution:** 3 good
**Rating:** 8
**Confidence:** 2

**Summary:**

In this paper the authors propose a new nested sampling algorithm based on Hamiltonian slice sampling. Their sampling method removes the linear dependence of the number of live points on dimensionality. They show that their algorithm runs significantly faster using parallelization in state-of-the-art programming frameworks. Empirically they show that their algorithm can scale up to higher dimensional problems compared to prior work. In addition, they show potential integration into the generative flow networks.

**Strengths:**

I think it is an interesting and novel idea that they combine the learning-based samplers with nested sampling algorithms. In this sense this paper is well motivated. The paper is also clearly written and the math derivations are sound, to my best knowledge. They also provide comprehensive evaluations on various tasks and show that their sampling algorithm can scale to higher-dimensional problems.

**Weaknesses:**

1. In the introduction section, the authors list 4 differences from prior work. Conceptually I am bit confused about how each of these 4 parts work with each other. I would hope that the authors can elaborate a bit.

2. A minor formatting issue: the caption of Figure 1 got cluttered and needs to be fixed later.

**Questions:**

Please see my questions in the previous section.

---

> ### Author Response · Authors · 2023-11-20
> **Response to Reviewer vEK5**
>
> We thank the reviewer for their encouraging feedback.
>
>
> To address the first point, we have added an appendix (Appendix H) with a complete GGNS algorithm. We have also added an ablation study (Appendix G) to analyse how our different improvements affect the algorithm’s performance.
>
>
> We have also corrected the formatting issue with the caption of Figure 1.

---

### Official Review · Reviewer_NqDj · 2023-10-31

**Soundness:** 3 good
**Presentation:** 3 good
**Contribution:** 3 good
**Rating:** 6
**Confidence:** 2

**Summary:**

The paper introduces Gradient-Guided Nested Sampling (GGNS), a novel nested sampling algorithm that utilizes Hamiltonian Slice Sampling and gradient information for improved scalability and efficiency, especially in high-dimensional settings. By leveraging differentiable programming frameworks and parallelization, GGNS achieves significant speed advancements, overcoming the dimensionality dependence that hampers previous methods. The paper also demonstrates GGNS's ability to integrate with generative flow networks for effective sampling from complex posterior distributions.

**Strengths:**

The paper is well structured and presents comprehensive experiments that highlight the merits of the proposed method. Notably, the method exhibits linear scaling with dimensionality, a feature not evident in prior techniques. Moreover, it outperforms existing approaches on several benchmark posterior sampling datasets. Furthermore, the authors present the idea clearly with the help of good visualizations.

**Weaknesses:**

My primary concern with the paper is the lack of clarity regarding the proposed method. While the contributions section seems to align with the methods, it primarily outlines algorithmic tweaks to existing methods. These minor modifications collectively seem to yield an impact. However, the paper does not showcase the final algorithm that encapsulates these changes; I only found references to current algorithms in the appendix. This absence of a clear differentiation in algorithmic terms hinders my ability to give a higher score, even though the experiments are commendable. Additionally, I found section 5 ambiguous, especially concerning the objectives of the experiments outlined there.

**Questions:**

-	I didn’t understand what was the advantage of the method for section 5, it seems a nice application that you can apply to generative flow networks, but what have we gained in training of the drift with your method, is there a baseline we can compare your method against?
-	Page 5 at the top there is some overlapping text below Figure 1.
-	I don’t really understand how you took advantage of ‘GPU interoperability’?
-	I don’t quite understand why there error bars reduce with the number of dimensions in Figure 3.

---

> ### Author Response · Authors · 2023-11-20
> **Response to Reviewer NqDj**
>
> We thank the reviewer for their very helpful feedback. We have tried our best to address these points with changes to the text.
>
>
> > lack of clarity regarding the proposed method
>
>
> We appreciate the difficulty in understanding how all the components of the algorithm come together. We have now added an appendix with a concrete GGNS algorithm (Appendix H).
>
>
> > I didn’t understand what was the advantage of the method for Section 5, it seems a nice application that you can apply to generative flow networks, but what have we gained in training of the drift with your method, is there a baseline we can compare your method against?
>
> In summary, Nested Sampling provides only a limited number of samples, but by training a GFlowNet, we can get an arbitrarily large number of samples by amortization.
>
> GFlowNets are off-policy deep reinforcement learning algorithms. They are trained to sample a target density, but work best when the training is guided by a dataset of meaningful ground truth samples (to aid mode discovery), which need not be large or unbiased. The aim of Section 5 is thus twofold: we show that
> - By using samples from GGNS in the training policy of a GFlowNet, we promote rapid mode discovery and convergence;
> - After the GFlowNet has been trained (with GGNS guidance), we are able to use the trained model to obtain nearly-unbiased samples in a fixed number of sampling steps.
>
> > Page 5 at the top there is some overlapping text below Figure 1.
>
> We have corrected the formatting error.
>
> > I don’t really understand how you took advantage of ‘GPU interoperability’?
>
> We have tried to clarify this point. The point here is that while Nested Samping algorithms had been written in differentiable programming languages before, none of those implementations took advantage of the gradient of the likelihood – their main advantage over implementations in non-differentiable programming languages arises from GPU interoperability. Therefore, what we meant to say (and we have tried to clarify), is that we do not take advantage only of GPU interoperability, but also of efficient parallel gradient computation.
>
> > I don’t quite understand why there error bars reduce with the number of dimensions in Figure 3.
>
> This is a very good question, which we had not fully explored in the original submission. The reason for this is that $\mathcal{D}_{\rm KL} (\mathcal{P} | \Pi)$ decreases as we increase the dimensionality, as we show in appendix F.2, meaning that the information gain when going from prior to posterior goes down. This could occur because of the cancellations between sine and cosine times, as the number of terms in the sum increases.

---

> > ### Comment · Reviewer_NqDj · 2023-11-22
> > **Response to rebuttal**
> >
> > I think the authors for taking time to address some of my concerns, I however do not feel it fundamentally changes my opinion so will keep my score.

---

### Official Review · Reviewer_ZsRi · 2023-11-01

**Soundness:** 2 fair
**Presentation:** 3 good
**Contribution:** 2 fair
**Rating:** 5
**Confidence:** 3

**Summary:**

This paper proposes a way to effectively combine Hamiltonian slice sampling with nested sampling.

**Strengths:**

* Nested sampling is a widely used algorithm for model comparison in physics. Therefore, improving the scalability of nested sampling to higher dimensions is an important problem.
* The paper provides a guideline on how to implement Hamiltonian slice sampling with nested sampling. It is well known that MCMC algorithms are very sensitive to such implementation detail. Therefore, this paper tackles a worthwhile issue.

**Weaknesses:**

While the paper discussed how to implementat nested sampling, the exact details of the implementations are not self-contained within the paper. Therefore, it is difficult to actually grasp *how* the paper is proposing to implement these parts. I expect a paper of this type to be self-contained in terms of details. While the authors do provide code, (note that the reviewers are not expected to look at the supplementary material) the paper should explain the algorithmic details, provide insight, and contrast with previous implementations. The paper mostly relies on textual explanation, which lacks the required technical preciseness. Below are some specific examples:

* Section 3 "Adaptive Time Step Control": "This time step is adjusted dynamically ... increase or decrease ..." How is it adjusted exactly?
* Section 3 "Trajectory Preservation": What is exactly a "trajectory" here? It is the states of the Markov chain after multiple Markov chain transitions? Or the intermediate states of a Hamiltonian trajectory as in recycled MCMC methods [1]?
* Section 3 "Trajectory preservation": "select a new live point at random from the stored trajectories" how random? Uniformly at random? Or weighted resampling as in typical Hamiltonian Monte Carlo implementations [2,3]?
* Section 3 "Pruning Mechanism": "This mechanism significantly improves the computational efficiency" How/why does it improve the computational efficiency exactly?
* Section 3 "Differentiable Programming": How is differentiable programming used here? Why does it help?

Furthermore, for quantatitive/theoretical claims, quantatitive/theoretical evidence (rigorous if possible) is necessary. There are multiple claims that were not entirely obvious to me:

* Section 3 second paragraph: "the fact that gradients guide the path means one no longer requires $n_{\text{live}} \sim O(d)$": I'm not sure if this is obvious. Is there a proof for this statement?
* Section 3 "Mode Collapse Mitigation": "... preventing them from converging prematurely to a single model": Is there theoretical/empirical evidence for this?
* Section 3 "Robust Termination Criterion": "terminate ... has decreased by a predetermined fraction from its maximum value": What is the theoretical principle behind this termination criterion?
* Section 6: "gradient-guided nested sampling ... makes use of the power of ... parallelization for significant speed improvements": I couldn't find any empirical evidence on much this method takes advantage of parallelization. Did the authors measure the strong scaling of this method? Until how many cores does this scale? What is the efficiency?

Lastly, I found the experiments inconclusive both in terms of experimental design and the choice of baselines.
* The paper claims that "the ingredients in GGNS ... to significantly improve its performance in high-dimensional settings ..." but none of the experiments are necessarily high-dimensional in todays standard. For instance the synthetic experiments in Section 4.1 Figure 1 only go as high as 128. See [4] Section 4.5 where the dimensionality goes as high as tens of thousands. While the method could be said to be scalable among nested sampling algorithms only, one could then question the significance of making nested sampling more scalable where more scalable alternatives exist.
* The exact contribution of each design choices in Section 3 are not evaluated independently. Therefore, it is unclear how much each of the components are contributing to any performance improvement. Given that no theoretical evidence is provided, I would expect a thorough empirical analysis and motivations for the design decisions. A great example of this is the no-u-turn sampler paper [5], which provided two innovations: tuning the trajectory length and the stepsize. They provide separate evaluation for each: Figure 3,4 for tuning the stepsize and Figure 5 for the trajectory length.
* Furthermore, some of the baselines are unclear. In Section 4.2, the paper states that HMC was used as a baseline. But HMC alone does not produce an estimate for the log-evidence unless special tricks are used like the harmonic mean estimator or Chib's method. How was HMC used to produce a log-evidence exactly? Similarly, the paper cites Halton (1962) for sequential Monte Carlo, but this paper seems unrelated to the sequential Monte Carlo used for estimating log-evidences as in [6]. Was this the intended citation? People usually attribute the genesis of sequential Monte Carlo to the bootstrap particle sampler [7] or the later seminal works [8,9].
* Moreover, the baselines are insufficient to really judge the performance of the method. At least in statistics, log-evidence estimation is popularly done using thermodynamic integration or bridge sampling [10]. Also, if the authors did intend to compare against sequential Monte Carlo as in [6], more implementation details are needed to really judge its validity, since SMC is notable for being sensitive to implementation details.
* It is also curious why the authors did not use the same set of baselines for all problems. For instance, comparable nested sampling methods are only used in Section 4.1.

### Minor Comments
* Section 1 first paragraph: Hamiltonian Monte Carlo was initially developed by Duane et al. [11].
* A similar reflective version of the HMC algorithm was developed by [12], although they did not consider the nested sampling setting. The authors might be interested to take a look.

### References
I am not the author of nor affiliated with the authors of the following papers.
1. Nishimura, Akihiko, and David Dunson. "Recycling intermediate steps to improve Hamiltonian Monte Carlo." (2020): 1087-1108.
2. Neal, Radford M. "MCMC using Hamiltonian dynamics." Handbook of markov chain monte carlo 2.11 (2011): 2.
3. Betancourt, Michael. "A conceptual introduction to Hamiltonian Monte Carlo." arXiv preprint arXiv:1701.02434 (2017).
4. Buchholz, Alexander, Nicolas Chopin, and Pierre E. Jacob. "Adaptive tuning of hamiltonian monte carlo within sequential monte carlo." Bayesian Analysis 16.3 (2021): 745-771.
5. Hoffman, Matthew D., and Andrew Gelman. "The No-U-Turn sampler: adaptively setting path lengths in Hamiltonian Monte Carlo." J. Mach. Learn. Res. 15.1 (2014): 1593-1623.
6. Dai, Chenguang, et al. "An invitation to sequential Monte Carlo samplers." Journal of the American Statistical Association 117.539 (2022): 1587-1600.
7. Gordon, Neil J., David J. Salmond, and Adrian FM Smith. "Novel approach to nonlinear/non-Gaussian Bayesian state estimation." IEE proceedings F (radar and signal processing). Vol. 140. No. 2. IET Digital Library, 1993.
8. Chopin, Nicolas. "A sequential particle filter method for static models." Biometrika 89.3 (2002): 539-552.
9. Del Moral, Pierre, Arnaud Doucet, and Ajay Jasra. "Sequential monte carlo samplers." Journal of the Royal Statistical Society Series B: Statistical Methodology 68.3 (2006): 411-436.
10. Gelman, Andrew, and Xiao-Li Meng. "Simulating normalizing constants: From importance sampling to bridge sampling to path sampling." Statistical science (1998): 163-185.
11. Duane, Simon, et al. "Hybrid monte carlo." Physics letters B 195.2 (1987): 216-222.
12. Mohasel Afshar, Hadi, and Justin Domke. "Reflection, refraction, and hamiltonian monte carlo." Advances in neural information processing systems 28 (2015).

**Questions:**

* Figure 3 why are the error bars decreasing as the dimensionality increases? I would assume higher dimensions are more challenging and therefore more variance. Is it not the case? In fact, in Figure 1, which I presume is the same type of plot, the results do seem intuitive.
* Section 1 second paragraph: "From the perspective of differentiable programming, less attention has been paid in recent years" I did not quite understand the intention of this sentence. In what context does differentiable programming have something to do with sampling here?

---

> ### Author Response · Authors · 2023-11-20
> **Response to Reviewer ZsRi (1/2)**
>
> We thank the reviewer for their very helpful and through feedback. We have tried our best to address the reviewer’s concerns.
>
>
> > the exact details of the implementations are not self-contained within the paper
>
>
> First, we appreciate the comment about the lack of details of the algorithm. Therefore, we have added an Appendix H with the details of our implementation. We hope this will add clarity and transparency to the paper.
>
>
> > "Adaptive Time Step Control": "This time step is adjusted dynamically ... increase or decrease ..." How is it adjusted exactly?
>
>
> The details of how adaptive time step happens are now in lines 21-24 of Algorithm 3.
>
>
> > "Trajectory Preservation": What is exactly a "trajectory" here? It is the states of the Markov chain after multiple Markov chain transitions? Or the intermediate states of a Hamiltonian trajectory as in recycled MCMC methods [1]?
> > "Trajectory preservation": "select a new live point at random from the stored trajectories" how random? Uniformly at random? Or weighted resampling as in typical Hamiltonian Monte Carlo implementations [2,3]?
>
>
> By trajectory, we mean the steps of the Markov chain, we have clarified it in a footnote in Section 3. The points are sampled fully randomly, as we need to sample uniformly from the prior. We have also clarified this.
>
>
> > "Pruning Mechanism": "This mechanism significantly improves the computational efficiency" How/why does it improve the computational efficiency exactly?
>
>
> The pruning mechanism improves the computational efficiency because, without it, we would continue to evaluate the likelihood for points that have drifted far away from the region of interest, leading to a waste of computational resources. We have tried to clarify this.
>
>
> > "Differentiable Programming": How is differentiable programming used here? Why does it help?
>
>
> Differentiable programming is key here, in that our method requires calculating the gradient of the likelihood. Therefore, it will only work for likelihoods for which we can compute gradients
>
>
> > second paragraph: "the fact that gradients guide the path means one no longer requires ": I'm not sure if this is obvious. Is there a proof for this statement?
>
>
> We have added a intuitive argument for why we get a linear scaling with dimensionality for our method. The idea is that because the gradient is $SO(n)$-equivariant, HSS can explore the principal directions in a number of steps that is independent of the dimension of the ambient space. Please also see the answers to Reviewers skHM and Ec9X for intuition regarding this.
>
>
> > "Mode Collapse Mitigation": "... preventing them from converging prematurely to a single model": Is there theoretical/empirical evidence for this?
>
>
> We have added empirical evidence for this in our ablation study (Appendix G.3).
>
>
> > "Robust Termination Criterion": "has decreased by a predetermined fraction from its maximum value": What is the theoretical principle behind this termination criterion?
>
>
> We have added a more detailed motivation for our termination criterion in appendix I, as well as empirical evidence of it working better than the commonly used in in appendix G.4.
>
>
> > "gradient-guided nested sampling ... makes use of the power of ... parallelization for significant speed improvements": I couldn't find any empirical evidence on much this method takes advantage of parallelization. Did the authors measure the strong scaling of this method? Until how many cores does this scale? What is the efficiency?
>
>
> It is hard to do an empirical study of how much we benefit from parallelization, as it would require rewriting the algorithm in a non-parallel way. Our argument is that, because our nested sampling steps are all run in an embarrassingly parallel way, our algorithm will benefit from an increased number of cores, up to $n_{live} / 2$, which is the number of points we can update simultaneously.
>
>
> A more detailed study with different computing configurations would be interesting but beyond our capabilities for the response period. Beyond that, we cast our contribution, in part, as an adaptation of nested sampling algorithms to hardware intended for modern machine learning workflows, featuring massive parallelization on GPUs. This is particularly important in data processing settings that combine nested sampling with deep learning, such as when the prior or likelihood models are given by deep neural networks. We have added this to the paper.

---

> > ### Author Response · Authors · 2023-11-20
> > **Response to Reviewer ZsRi (2/2)**
> >
> > > The paper claims that "the ingredients in GGNS ... to significantly improve its performance in high-dimensional settings ..." (...). While the method could be said to be scalable among nested sampling algorithms only, one could then question the significance of making nested sampling more scalable where more scalable alternatives exist.
> >
> > We tested the algorithm up to $d=256$, which is a higher dimensionality that has ever been used in nested sampling. In such high dimensions, simpler MCMC-based samplers (HMC, NUTS) have traditionally been used. Our proposed approach is the first to bring the benefits of nested sampling – namely, the ability to reliably sample multimodal distributions and calculate the evidence – to such high-dimensional spaces.
> >
> > > The exact contribution of each design choices in Section 3 are not evaluated independently. Therefore, it is unclear how much each of the components are contributing to any performance improvement.
> >
> > Appendix G provides a detailed ablation study, focusing precisely on that. See the text, as well as the response to all reviewers, for discussion.
> >
> > > Furthermore, some of the baselines are unclear. In Section 4.2, the paper states that HMC was used as a baseline. But HMC alone does not produce an estimate for the log-evidence unless special tricks are used like the harmonic mean estimator or Chib's method. How was HMC used to produce a log-evidence exactly?
> >
> > This result is taken from [Zhang and Chen, 2022], which presumably used the harmonic mean estimator to calculate the evidence given the obtained samples.
> >
> > > The paper cites Halton (1962) for sequential Monte Carlo, but this paper seems unrelated to the sequential Monte Carlo used for estimating log-evidences as in [6]. Was this the intended citation? People usually attribute the genesis of sequential Monte Carlo to the bootstrap particle sampler [7] or the later seminal works [8,9].
> >
> > We thank the reviewer pointing out this oversight, we have now added these citations to the paper.
> >
> > > Moreover, the baselines are insufficient to really judge the performance of the method. At least in statistics, log-evidence estimation is popularly done using thermodynamic integration or bridge sampling [10].
> >
> > The estimation of the log-evidence is only one of the benchmarks we use, and it is commonly used to benchmark sampling methods. We used this benchmark because it has also been used in [Zhang and Chen, 2022] and [Lahlou et al., 2023].
> >
> >
> > > Also, if the authors did intend to compare against sequential Monte Carlo as in [6], more implementation details are needed to really judge its validity, since SMC is notable for being sensitive to implementation details.
> >
> >
> > We are using the result taken from [Zhang and Chen, 2022]. We have highlighted their SMC implementation details in the paper, for clarity.
> >
> >
> > > It is also curious why the authors did not use the same set of baselines for all problems. For instance, comparable nested sampling methods are only used in Section 4.1.
> >
> >
> > The logic here was that Section 4.1 shows our nested sampling implementation outperforms others, so the next sections focus on comparing us to different methods. The amount of available methods is so large, that it felt very impractical to compare with every method on every experiment.
> >
> >
> > > Section 1 first paragraph: Hamiltonian Monte Carlo was initially developed by Duane et al. [11].
> >
> >
> > We thank the referee for pointing this out. We have added the citation.
> >
> >
> > > A similar reflective version of the HMC algorithm was developed by [12], although they did not consider the nested sampling setting. The authors might be interested to take a look.
> >
> > This is certainly a very relevant paper, which we were not aware of. We have now added a citation to the paper, as well as placed it in the context of our work.
> >
> >
> > > Section 1 second paragraph: "From the perspective of differentiable programming, less attention has been paid in recent years" I did not quite understand the intention of this sentence. In what context does differentiable programming have something to do with sampling here?
> >
> >
> > What we meant here was that differentiable programming, and particularly differentiable likelihoods, has been used for sampling methods such as HMC, SMC, as well as deep learning methods. But not for nested sampling.

---

> ### Comment · Reviewer_ZsRi · 2023-11-21
> **Response**
>
> I sincerely thank the authors for addressing my comments. In fact, I am impressed by the effort they have put in. Unfortunately, I have some major concerns that remain, that prevent me from increasing my score. In particular, one of the main claims of the paper that the method is scalable in terms of dimensionality, is still not clearly demonstrated.
>
> * **Pooled Baselines?**
>  For objectively judging the performance in a controlled manner, I believe the experiments producing Figure 1 and Figure 2 are the most important. However, the baselines are "pooled" and not properly evaluated on the same problems. While the authors replied that, after showing that their method performs well on one experiment, they chose to "move on" to a different set of baselines, I do not quite agree that this is scientifically rigorous. Is it obvious that we will see the same experimental result despite a different setup?
> * **Lack of High-Dimensional Experiment**
> Furthermore, the experiment for Figure 2 is way too weak. For instance, take a look at Table 1 by Zhang and Chen (2022). There, they included a Log-Gaussian Cox process experiment, which is both geometrically challenging and also actually high-dimensional. On the other hand, the current paper only considers the 10-D funnel and a 2-D mixture. The remaining experiments have a different set of baselines for some reason, which, again, I do not find that it provides clear evidence of superiority.
> * **Paper Organization**
> Lastly, while I appreciate the authors' effort to improve the paper and incorporate the comments of the reviewers, the paper is still not self-contained. This unfortunately, does not address the concern that other reviewers have also raised. I expect some major reorganization to make the paper meet the bar for the quality expected for an ICLR paper.
>
> Lastly, I have some additional minor comments. The paper states, "Differentiable Programming Whilst nested sampling algorithms written in differential programming languages exist," for which the authors clarified the meaning. I believe this wording is quite misleading. The sampler *itself* does not *internally* use differentiable programming. (There are, in fact, samplers that internally use differentiable programming in a non-trivial manner, for example, by differentiating through the Matropolis-Hastings correction step. So it's worth clarifying that the method is not doing something like that here.) All that it operates with, are the provided gradients right? I think it would be worth clarifying the language that the proposed methodology takes advantage of *gradients* not necessarily differentiable programming.  Furthermore, for the experiment in Figure 2, the authors mention that what is called "HMC" here used the harmonic mean estimator following Zhang and Chen. Unfortunately, the harmonic mean estimator is known as "the worst Monte Carlo estimator ever" due to its potentially infinite variance. Therefore, it cannot be considered to be a valid baseline for estimating marginal likelihoods. I recommend the authors look for better baselines, for instance, thermodynamic integration.

---

> > ### Author Response · Authors · 2023-11-23
> > **Response**
> >
> > We thank the reviewer for taking the time to get back to us:
> >
> > > Pooled baselines
> >
> > We do not fully understand what the reviewer means here by “pooled”. Figure 1 shows that our algorithm can be scaled to higher dimensional problems at a fixed number of live points, unlike competing NS algorithms. We have also added theoretical arguments for this improved scaling in section 3.
> >
> > > Lack of High-Dimensional Experiment
> >
> > To add a higher dimensional experiment, we just submitted a job redoing Fig. 2 with 50x50 and 100x100 images, though we are not sure it will finish before the discussion period is over. We will add these to the final version of the paper.
> >
> > > Paper Organization
> >
> > We are confused about this point, as we feel like we have addressed the points raised by the other reviewers in this regard. The paper now contains an implementation with all the algorithm details, an ablation study, and theoretical arguments for improved scaling. Therefore, we are not sure what the paper is lacking in terms of reorganization.
> >
> > >  Therefore, it cannot be considered to be a valid baseline for estimating marginal likelihoods. I recommend the authors look for better baselines, for instance, thermodynamic integration.
> >
> > The reason we chose this baseline is that it has been used by works such as Zhang and Chen 2021; Lahlou et al. 2023, which are considered SOTA in similar problems. Therefore, it was our understanding that our improved performance in the same baselines would provide a strong argument in favour of our method.

---

> ### Comment · Reviewer_ZsRi · 2023-11-23
>
> Dear Authors,
>
> > > Paper Organization
> >
> > We are confused about this point, as we feel like we have addressed the points raised by the other reviewers in this regard. The paper now contains an implementation with all the algorithm details, an ablation study, and theoretical arguments for improved scaling. Therefore, we are not sure what the paper is lacking in terms of reorganization.
>
> I am specifically referring to my original comment about "self-containment." By self-containment, I am saying that the details currently deferred to the appendix should have been blended in the main text. I believe readers should be able to have a rough idea of the proposed algorithm after having read only the main text, which does not seem to be the case. Recall that other reviewers have also expressed similar concerns, and I feel the main text still does not properly address this. Furthermore, given the current organization of the paper, it appears to me that this will require a major restructuring of the paper, which would require an additional round of thorough review, unfortunately.
>
> Note that all of this is for the benefit of the paper, I strongly believe the technical contribution of the paper deserves better presentation.
>
> Nevertheless, given many of my original concerns have been addressed, I'll raise my score.

---

> > ### Author Response · Authors · 2023-11-23
> > **Response**
> >
> > Thank you again for your time in engaging with our paper and for the very detailed feedback, which we believe has helped strengthen the paper.

---

### Official Review · Reviewer_Ec9X · 2023-11-03

**Soundness:** 2 fair
**Presentation:** 2 fair
**Contribution:** 2 fair
**Rating:** 5
**Confidence:** 3

**Summary:**

The authors propose a general-purpose gradient-guided nested sampling algorithm, GGNS, combining the state of the art in differentiable programming, Hamiltonian slice sampling, clustering, mode separation, dynamic nested sampling, and parallelization.  The authors show that the combination leads to faster mode discovery and more accurate estimates of the partition function.

**Strengths:**

The authors present a comprehensive review of related works.

**Weaknesses:**

1. The main contribution of the paper is a combination of different existing methods. The paper lacks originality and significance.

3. The reviewer cannot find the proposed GGNS  algorithm. The authors need to give a concrete GGNS  algorithm.

2. There is no theoretical analysis regarding the GGNS. There is no theoretical guarantee for the advantages of the GGNS algorithm.

4. Since the method is a combination of different strategies, the authors need to conduct an abolition study on the strategies to validate their effectiveness.

**Questions:**

See weaknesses.

---

> ### Author Response · Authors · 2023-11-20
> **Response to Reviewer Ec9X**
>
> We thank the reviewer for their feedback. We have tried to address all four points that were brought up:
>
> 1. We have tried to further clarify our main contributions. Please see response to all reviewers for a summary.
>
>
> 2. We thank the reviewer for bringing up this point. We have now added an appendix (Appendix H) with a concrete GGNS algorithm, including all the details.
>
>
> 3. We appreciate this point. To attempt to address it, we have tried to provide a justification for what is, in our opinion, the most important part of this work, which is the fact that each step only needs O(1) bounces; as this is what allows the algorithm to have a better scaling with the number of dimensions. This has been added to Section 3. In summary, while slice sampling only has information about one dimension per step (needing $O(n)$ steps to explore an $n$-dimensional space), when we use Hamiltonian slice sampling, the gradient provides information about all $n$ dimensions on each step. The $SO(n)$-equivariance of Hamiltonian slice sampling implies that exploration of a density concentrated around a low-dimensional manifold is less sensitive to the dimension of the ambient space. Please also see the answer to Reviewer skHM for intuition regarding this.
>
>
> 4. This is a very good point. We have added an Appendix G with a detailed ablation study. Please see the response to all reviewers for discussion.

---

> > ### Comment · Reviewer_Ec9X · 2023-11-23
> >
> > Thank the authors for the clarification and improvements. However, the main contribution is still a combination of different strategies, and it is limited.  I would like to increase the score to 5.

---

### Official Review · Reviewer_skHM · 2023-11-09

**Soundness:** 3 good
**Presentation:** 2 fair
**Contribution:** 3 good
**Rating:** 8
**Confidence:** 3

**Summary:**

This paper introduces a new nested sampling algorithm which
through a pruning method and adaptive step size results in
asymptotically fewer likelihood evaluations compared to the
current state of the art. Experiments show that the method
in addition to being more computational efficient yields
higher quality samples as compared to popular libraries
doing the same thing

**Strengths:**

This is highly original work that makes a significant contribution
to the literature on nested sampling. The method has compelling experiments
to support it, and the underlying techniques are reasonably well-explained.

**Weaknesses:**

As the paper includes many contributions, it would have been nice to see
an ablation study showing how each of the contributions work to improve
the nested sampler. Additionally, some contributions like Mode Collapse
Mitigation only have supporting evidence in the appendix. It would be nice
if some of those figures could be in the main paper.

An additional concern I have relates to the main claim that the sampler
only need O(1) bounces. I saw no proofs or even intuition for why this is
the case. For something so core to the paper, it would greatly improve the
paper if even some intuition was provided for why that might be the case.

Minor Issues:

There are a few typoes and misspellings that should be fixed.

**Questions:**

What suggests only O(1) bounces are needed?
Would it be possible to do an ablation study?

---

> ### Author Response · Authors · 2023-11-20
> **Response to Reviewer skHM**
>
> We thank the reviewer for their very helpful feedback and comments. We particularly appreciate the suggestion of an ablation study. We have now added an Appendix G with such a study. Please see the response to all reviewers for discussion.
>
>
> With respect to the $O(1)$ bounces, the reviewer makes a good point that our original submission did not justify this claim. We have now added a theoretical justification for this point in Section 3. The main idea is that, while other methods like slice sampling only have information about one dimension per step (needing $O(n)$ steps to explore an $n$-dimensional space), when we use Hamiltonian slice sampling, the gradient provides information about all $n$ dimensions on each step, meaning we need only $O(1)$ steps.
>
>
> For intuition, consider a density concentrated around a $d$-dimensional linear subspace of an $n$-dimensional space ($d\ll n$). Usual (coordinatewise) slice sampling would require $O(n)$ bounces to mix within the subspace despite the space having only $d$ degrees of freedom. On the other hand, Hamiltonian slice sampling, which is equivariant to orthogonal transformations of the space, would take steps within the low-dimensional subspace and does not depend on the dimension of the ambient space.
>
>
> We have also fixed multiple typos and misspellings.

---

> > ### Comment · Reviewer_skHM · 2023-11-22
> >
> > Yes, I think this greatly improves the paper! Thank you for the changes. I will update my score accordingly!

---

> > > ### Author Response · Authors · 2023-11-23
> > > **Response**
> > >
> > > We thank you for your kind words… We wonder if you have perhaps forgotten to update the score. Thank you again!

---

### Author Response · Authors · 2023-11-20
**Response to all reviewers + new results**

We thank all the reviewers for their comments on our paper. Motivated by your suggestions, we have updated the manuscript with several changes, notably:
- **We have added a complete description of the GGNS algorithm in Appendix H.**
- **We have added ablation studies that isolate the effect of each component of the proposed algorithm in Appendix G.** The main conclusion from the ablation study is that by removing different parts of the GGNS algorithm, such as adaptive step, or trajectory preservation, the algorithm gets biased results on high dimensions. Our understanding is that for any given problem, there will be certain hyperparameters leading to unbiased sampling, but our algorithm has the capability to self-tune these parameters, leading to unbiased inference in various problems and dimensions.
- A result comparing estimation error with different number of likelihood evaluations has also been added (Appendix J).
- We have added discussion to Section 3 to motivate some choices in the proposed algorithm, and theoretical arguments for our improved dimensionality scaling.
- We have also tried address comments regarding the contributions section. In summary, our main contributions are:
  - **Combining, for the first time, ideas in nested sampling and Hamiltonian Monte Carlo**, to achieve a performance in sampling tasks that beats all existing methods. While we agree that the methods existed, they had never before been combined in a unified framework.
  - **Adapting nested sampling algorithms to hardware designed for modern machine learning workflows.** This includes utilizing GPUs for massive parallelization. This is especially crucial in data processing settings that involve nested sampling and deep learning, particularly when the prior or likelihood models are based on deep neural networks.
  - **The combination of Nested Sampling with GFlowNets** to generate a large number of samples from the target distribution. Again, this combination had never been used before, and it allows to train GFlowNets in problems where the existing training strategies would fail, as described in Section 5.
- Various other changes are described in the individual responses below.

We hope that our responses have satisfactorily addressed your questions and concerns. Please do not hesitate to let us know if we can provide further clarification.

---

> ### Author Response · Authors · 2023-11-20
> **New pdf posted**
>
> We have just uploaded a revised pdf with changes marked in $\color{red}\text{\sf red}$.

---

### Meta-Review · Area_Chair_5XcP · 2023-12-08

**Metareview:**

This paper proposes a gradient-guided sampling algorithm to explore complex high-dimensional distributions. This is achieved by combining Hamiltonian slice sampling with nested sampling.

High-dimensional sampling is an important problem, and providing implementation details for improved MCMC algorithms is a highly relevant topic. The weakness of the paper is in the description of the method, that is, there is no "main method" lucidly presented in the main text, and only added later to the appendix for a revised version.   Larger changes to the paper would be required to add clarity and details describing the proposed algorithm, and such a rework will benefit from another round of reviews.  Finally, the evaluations of the main claims and performance could be done more thoroughly, for example considering larger standard benchmarks.

I recommend to reject the paper and encourage the authors to take the reviewers' detailed feedbacks into account for a resubmission.

**Justification For Why Not Higher Score:**

The paper will benefit from a rewrite as lined out in the meta-review to clarify the method and its specific details.  Integrating such larger changes and feedbacks as given by the reviewers will substantially change the paper, and require another round of reviews.

**Justification For Why Not Lower Score:**

N/A

---

### Decision · Program_Chairs · 2024-01-16

Reject